# IneqSearch: Hybrid Reasoning for Olympiad Inequality Proofs

**Zhaoqun Li**[*]
Zhejiang University
Beijing Academy of Artificial Intelligence
lizhaoqun@zju.edu.cn

**Beishui Liao**[†]
Zhejiang University
baiseliao@zju.edu.cn

**Qiwei Ye**[†]
Beijing Academy of Artificial Intelligence
qwye@baai.ac.cn

## Abstract

Mathematicians have long employed decomposition techniques to prove inequalities, yet automating this process remains a significant challenge in computational mathematics. We introduce IneqSearch, a hybrid reasoning system that integrates symbolic computation with large language models (LLMs) to address this challenge. IneqSearch reformulates inequality proving as a structured search problem: identifying appropriate combinations of theorems that decompose expressions into non-negative components. The system combines a symbolic solver for deductive reasoning with an LLM-based agent for constructive proof exploration, effectively implementing methodologies observed in formal mathematical practice. A key contribution of IneqSearch is its iterative learning mechanism that systematically incorporates newly proven results into its theorem database, enabling knowledge acquisition during practice that enhances its capabilities without requiring human intervention. In empirical evaluation on 437 Olympiad-level inequalities, IneqSearch successfully proves 342 problems, significantly outperforming existing methods and demonstrating the effectiveness of integrating symbolic and neural approaches for mathematical reasoning.

## 1 Introduction

Automating the proof of Olympiad-level inequalities in computational mathematics represents a fundamental benchmark for evaluating the capabilities and boundaries of artificial general intelligence [1, 2, 3, 4]. Inequality proofs typically require both deductive inference and insightful transformations, making them valuable benchmarks for evaluating progress toward general-purpose mathematical reasoning. Recent approaches have made progress in mathematical reasoning through natural language proof generation using large language models (LLMs) [5, 6], formal proof techniques via automated theorem proving (ATP) [7, 8, 9, 10], and mixed methods that combine these two strategies [11, 12]. However, algebraic inequalities remain challenging, as they require constructing proofs within an infinite reasoning space.

To prove an inequality $F \geq 0$, a common and effective strategy is to transform $F$ into a more manageable form by decomposing the original expression into tractable components. For instance, many proofs in Olympiad inequalities rely on intricate scaling techniques, which often employ

---

[*]This work was done during an internship at BAAI

[†]Corresponding author

auxiliary lemmas (e.g., Cauchy-Schwarz) or identity-based rewritings. These techniques naturally lead to a decomposition of $F$, allowing it to be expressed as a sum of non-negative components, namely $F = F_1 + F_2 + \cdots$, where each $F_i \geq 0$. Inspired by this idea, we develop an automated approach that reformulates the proof of inequalities as a search problem, aiming to decompose the target expression $F$ into a sum of known non-negative terms, *i.e.*, $F = \sum_i F_i$. Implementing this strategy in practice faces two critical challenges: (1) Inequality decomposition requires strong deductive reasoning to manipulate complex expressions. (2) Constructive proofs demand innovative strategies to adapt lemmas to specific problem structures.

To address these challenges, we introduce IneqSearch, a hybrid reasoning system that integrates symbolic computation with large language models, in which inequality proving is framed as an iterative search process. Specifically, leveraging the robust deductive capability of symbolic computation, we introduce a symbolic solver that formulates inequality proving as a search problem, enabling precise and systematic deductive reasoning. In parallel, an LLM-based inductive agent is designed to explore constructive proofs, utilizing the inductive strengths of LLM to propose novel scaling techniques and transformations. A distinguishing feature of IneqSearch is its iterative learning mechanism, which autonomously incorporates solved inequalities into its knowledge base, continuously enhancing its proving capability and *scalability* to complex problems. This design reflects the cumulative nature of mathematical knowledge, where established results facilitate subsequent discoveries. Furthermore, IneqSearch ensures soundness by constructing verifiable proofs, and achieves *interpretability* through human-readable and transparent proving steps.

IneqSearch is evaluated on a benchmark of 437 Olympiad-level inequalities, significantly outperforming leading methods. Notably, on the challenging MO-INT dataset [1], our method achieves 100% success rate, demonstrating its effectiveness for particularly difficult cases.

## 2    Inequality Proving

We first formalize the inequality proving problem before introducing IneqSearch.

### 2.1    Problem Formulation

Formally, an inequality proving problem involves verifying non-negativity of an expression under specific conditions. For an inequality with $n$ variables $\{x_i\}_{i=1}^n \in \mathbb{R}^n$, a problem $P = \langle F, C \rangle$ is defined by an inequality expression $F(x_1, \ldots, x_n)$ and a condition expression $C(x_1, \ldots, x_n)$, which can be formulated as:

$$\text{Prove that } F(x_1, \ldots, x_n) \geq 0 \text{ when } C(x_1, \ldots, x_n) \bowtie 0, \tag{1}$$

where $\bowtie \in \{=, \geq, >\}$. We refer to $F$ as the inequality and $C$ as the assumption, where $C$ may be absent if no additional condition is required. To focus on core mathematical proof techniques, we assume all variables are positive, as cases involving negative values can typically be proven in a similar manner.

### 2.2    Decomposition-based Proving

The fundamental methodology underlying our approach reformulates inequality proving as a decomposition task [13]. This approach is based on the principle that many complex inequalities can be systematically reduced to combinations of established non-negative expressions. We employ two methods for inequality decomposition (see Appendix A for technical details):

**Elementary Cyclic Polynomial Decomposition (ECPD)**    ECPD provides a systematic framework for representing and manipulating expressions in inequality proofs. An elementary cyclic polynomial of degree $d$ in $n$ variables can be expressed as

$$C_{k_1, k_2, \ldots, k_m}^{(n)} = \sum_{cyc} x_1^{k_1} x_2^{k_2} \cdots x_m^{k_m},$$

where $m \leq n$ and $k_1 + k_2 + \cdots + k_m = d$. This representation enables the transformation of complex expressions into standardized forms, making them more amenable to systematic analysis. For example, we have:

$$(n-1)\left(C_1^{(n)}\right)^2 - 2nC_{1,1}^{(n)} \geq 0, \quad \left(C_1^{(3)}\right)^3 - 2C_1^{(3)}C_{1,1}^{(3)} - 3C_{2,1}^{(3)} \geq 0.$$

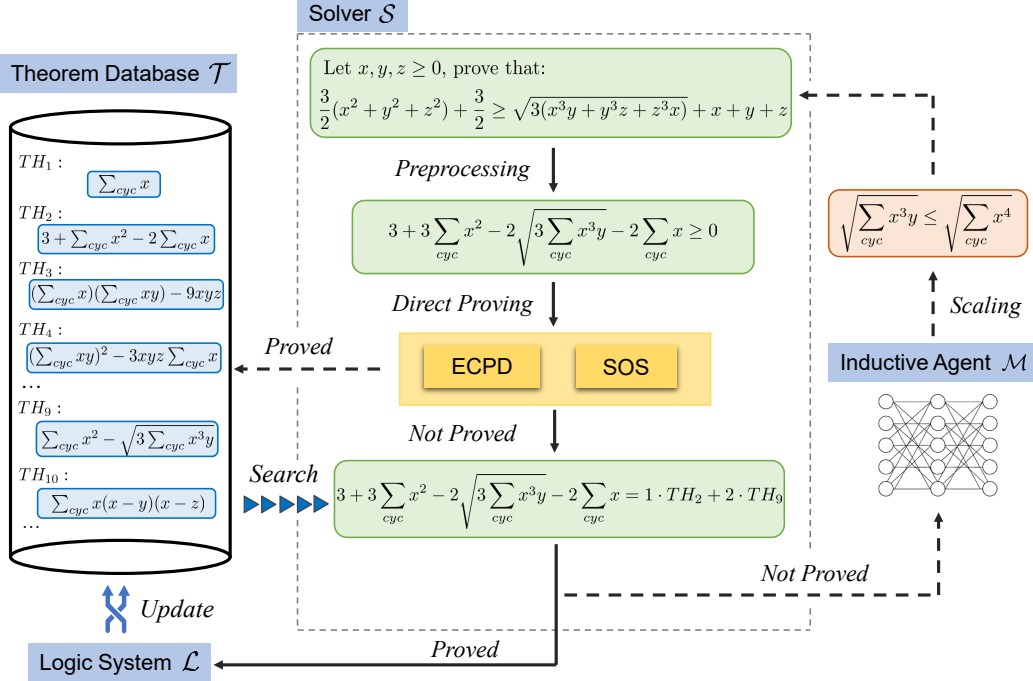

Figure 1: Overview of the proving pipeline. The solver starts proving by processing the given problem into a standard format. Then theorem search identifies applicable theorems from the database and attempts to decompose the inequality. If the proof fails, an inductive agent explores transformations to generate potential directions. If the proof succeeds, the logic system expands the theorem database by integrating newly proven results.

**Sum of Squares (SOS)**   The SOS method provides a powerful certification mechanism for polynomial non-negativity through quadratic decomposition. A polynomial $T(x_1, \ldots, x_n)$ is an SOS polynomial if it can be expressed as $T = \sum_i t_i^2$, where each $t_i$ is a real polynomial. For example, $x^4 + y^4 - 4xy + 2 = (x^2 - y^2)^2 + 2(xy - 1)^2$ is a sum of squares. In practice, finding an SOS decomposition is formulated as a semidefinite programming problem, enabling automated discovery of decompositions for complex polynomials.

## 3   The IneqSearch System

The development of IneqSearch is motivated by the observation that effective inequality proving requires both systematic deductive reasoning and constructive insight. The system architecture reflects this dual requirement, implementing a methodology that progresses from direct proof attempts to theorem-based search, and finally to exploration of problem transformations when simpler approaches are insufficient. As depicted in Figure 1, IneqSearch is built upon a proving pipeline that integrates symbolic computation, theorem search, and language model-guided exploration.

### 3.1   The Symbolic Solver

The symbolic solver $\mathcal{S}$ forms the deductive core of IneqSearch, enabling robust algebraic manipulations through preprocessing and theorem search.

**Preprocessing**   Given a problem $P = \langle F, C \rangle$, the solver first standardizes it to facilitate subsequent processing: (1) For homogeneous inequalities without explicit constraints, $\mathcal{S}$ introduces an additional normalization assumption $\sum_i x_i = 1$. (2) Fractional expressions are converted to equivalent numerator-only forms through algebraic manipulation. (3) The solver attempts to eliminate radical

subexpressions in $F$ using techniques analogous to `unrad` in SymPy; if unsuccessful, it retains the original form.

The preprocessing step ensures that the original problem is systematically transformed into an equivalent and standardized form, making it suitable for subsequent procedures.

**Theorem Search**  For simple problems, direct application of ECPD or SOS methods often suffices. When these approaches are inadequate, IneqSearch implements a more comprehensive search strategy. This search systematically examines possible representations of the target expression $F$ as a linear combination of established theorems from the database, effectively decomposing the original problem into previously solved components.

The theorem database $\mathcal{T}$ stores axioms and proven theorems in the form $t_i = \langle g_i, h_i \rangle$, indicating that $g_i \geq 0$ under the assumption $h_i = 0$. A direct corollary is that any non-negative linear combination of theorems is also non-negative:

$$\sum_i \lambda_i g_i \geq 0, \quad \forall i,\, h_i = 0 \wedge \lambda_i \geq 0. \tag{2}$$

Given an inequality expression, the solver identifies applicable theorems, *i.e.* those for which $h_i = 0$ is satisfied, and formulates the search problem as follows:

$$Find\ \lambda_i\ and\ f_i\ such\ that$$
$$F = \sum_i^k \lambda_i f_i, \quad \lambda_i \geq 0, \quad f_i \in \mathcal{T}_c, \tag{3}$$

where $\mathcal{T}_c = \{g_i \mid i = 1, 2, \ldots, k\}$ is the set of applicable theorems. The search problem corresponds to solving a system of equations for the coefficients $\{\lambda_i\}_{i=1}^k$. To obtain a numerical solution, we formulate a linear programming problem to determine feasible real values for $\{\lambda_i\}_{i=1}^k$. Specifically, if the search problem is feasible, $\mathcal{S}$ tries to find a numerical solution where the coefficients $\{\lambda_i\}_{i=1}^k$ are non-negative and satisfy all the required constraints of the problem. When the entire search is successful, the inequality problem is proven; otherwise, the proof fails.

## 3.2   The Inductive Agent

For challenging problems where the symbolic solver fails, it may become necessary to rely on more sophisticated algebraic methods, including the adoption of widely used lemmas such as the Cauchy-Schwarz inequality. We employ an LLM-based inductive agent $\mathcal{M}$ to explore constructive proof strategies. The agent analyzes the structure of the inequality and proposes transformations using well-known inequalities (such as AM-GM, Cauchy-Schwarz, etc.) and homogenization, enabling further exploration. As illustrated in Figure 2, the exploration follows two directions: (1) Application of auxiliary inequalities to scale the original expression into more manageable forms. (2) Target shift by replacing constants based on given assumptions to achieve more homogeneous expressions.

For each difficult problem, given the infinite possibilities of constructive approaches, we ask $\mathcal{M}$ to provide multiple possible choices at each step, and let $\mathcal{S}$ verify the validity of the explored scaling techniques.

**Proof Construction Example**  Taking the case in Figure 2 as an example, the solution process begins with $\mathcal{S}$ attempting direct methods, finding that neither ECPD nor SOS produces an immediate result. The system then queries its theorem database but fails to identify a suitable combination of existing results. Subsequently, the inductive agent analyzes the inequality structure and proposes several choices.

For the left-hand side expression, $\mathcal{M}$ considers it suitable to adopt the Cauchy-Schwarz inequality, which states that $(\sum a^2)(\sum b^2) \geq (\sum ab)^2$. $\mathcal{M}$ then proposes two possible constructions by assigning $b = \sum_{cyc} x^2 + yz$ and $b = \sum_{cyc} x(x^2 + yz)$. For the right-hand side, which is a constant, it can be rewritten using the assumption, enabling the original problem to be reformulated in several equivalent forms. For example, $\mathcal{M}$ multiplies it by $\sum_{cyc} x^2/3$ or $\sqrt{\sum_{cyc} x^2/3}$. Using these transformations, the symbolic solver can attempt to prove a total of 8 additional inequality

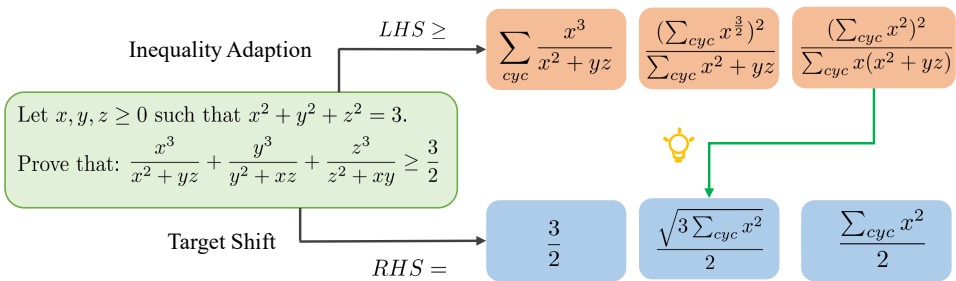

Figure 2: Exploration strategies utilizing the Cauchy-Schwarz inequality and homogenization. The exploration process proceeds in two directions: $\mathcal{M}$ adapts the inequality by transforming the left-hand side into multiple manageable forms, and rewrite the right-hand side into equivalent expressions that eliminate assumption.

forms. Finally, $\mathcal{S}$ successfully proves $\frac{(\sum_{cyc} x^2)^2}{\sum_{cyc} x(x^2+yz)} \geq \sqrt{3\sum_{cyc} x^2}/2$ under no assumptions. This completed proof is subsequently integrated into the theorem database, demonstrating how IneqSearch combines deductive verification with constructive exploration to expand its proving capabilities.

### 3.3 The Self-improving Mechanism

A distinguishing feature of IneqSearch is its iterative learning capability that continuously expands the theorem database. As discussed earlier, the proving capability of IneqSearch is directly tied to the richness of the theorem database $\mathcal{T}$. To enhance this capability, IneqSearch employs an iterative learning paradigm that continuously expands $\mathcal{T}$ by incorporating previously proven inequalities.

**Logic System $\mathcal{L}$** During the proving process of a problem $P$, $\mathcal{L}$ maintains a set $S_P$ containing all intermediate conclusions. After completion, proven theorems from $S_P$ are selectively incorporated into $\mathcal{T}$. For instance, if a proof forms a chain $E_1 \geq 0, E_2 \geq 0, \ldots, E_m \geq 0$ where $(E_i \geq 0) \vdash (E_{i+1} \geq 0)$, each $E_i$ is analyzed and potentially added to the database. This ensures that every theorem in $\mathcal{T}$ has been rigorously validated, guaranteeing the system's soundness.

**Iterative Learning** Initially, $\mathcal{T}$ contains only basic axioms (Appendix A.3). As IneqSearch solves problems, it progressively learns new theorems. This self-improving mechanism enables the system to tackle increasingly complex problems over time, as each solved problem potentially provides new tools for addressing more challenging cases.

### 3.4 Formal Proof Generation

```
Process:: To prove:
Inequality (1 + 1/x)*(1 + 1/y)*(1 + 1/z) - 64 >= 0, Assumption x + y + z - 1 = 0.

Simplify:: After simplification of numerator:
Inequality -64*x*y*z + (x + 1)*(y + 1)*(z + 1) >= 0, Assumption x + y + z - 1 = 0.

Transform:: Update after ECPD:
Σxy - 63*xyz + 2, Eq(Σx, 1).

Method:: Search existing theorem set.
Result:: Proved by theorem (TH_3) (TH_10):
Σxy - 63*xyz + 2 = (1)*(Σxy - 9*xyz) + (2)*(1 - 27*xyz) >= 0.
```

Figure 3: Proof generated by IneqSearch.

The overall proof logic is straightforward to understand. It primarily involves algebraic transformations and theorem search, systematically breaking down complex inequalities into manageable steps. The proof generation process mainly focuses on logging the procedure, including the transformations applied, the use of assumptions, and the theorems referenced during the proof. For instance, in

Table 1: Performance comparison of different models.

| Proof Type | Model | INEQ-437 | | MO-INT-20 | |
|---|---|---|---|---|---|
| | | # Solved | Success Rate | # Solved | Success Rate |
| Natural Language | Gemini-2.0-Flash | 62 | 14.2% | 0 | 0% |
| | DeepSeek-R1 | 149 | 34.1% | 4 | 20% |
| | OpenAI o3-mini | 154 | 35.2% | 3 | 15% |
| Lean4 | Gemini-2.0-Flash | 0 | 0% | 0 | 0% |
| | DeepSeek-Prover-V1.5 | 1 | 0.2% | 0 | 0% |
| | OpenAI o3-mini | 2 | 0.5% | 0 | 0% |
| | AIPS | – | – | 10 | 50% |
| | LIPS w/ 4o-mini | 264 | 60.4% | 18 | 90% |
| | LIPS w/ o3-mini | 273 | 62.5% | 18 | 90% |
| Formal | IneqSearch w/ 4o-mini | 334 | 76.4% | 19 | 95% |
| | IneqSearch w/ o3-mini | 342 | 78.3% | 20 | 100% |

Figure 3, each transformation step is logged, providing a clear and transparent explanation of the process.

## 4 Experimental Results

To evaluate the performance of our method, a comparison of IneqSearch with baseline and state-of-the-art methods is presented, along with a performance analysis of each component through an ablation study. The iterative progress is analyzed to observe the system's self-improving capabilities. In the end, several case studies are provided for the visualization of IneqSearch's advantages.

### 4.1 Experimental Setup

**Data Collection** We constructed INEQ-437, a benchmark of 437 Olympiad-level inequalities compiled from authoritative sources [14, 13]. We included all problems expressible in the form of Equation 1 without manual selection, representing approximately 90% of inequalities in these sources. The detailed content of the benchmark can be found in the supplementary materials.

**Baselines** We evaluated IneqSearch against leading LLMs (Gemini-2.0-Flash [15], OpenAI o3-mini [16], DeepSeek-Prover-V1.5 [17], DeepSeek-R1 [18]) and hybrid systems (AIPS [1], LIPS [19]) across both natural language and formal proofs. The temperature hyperparameter is set to 0.3 for all language models when applicable.

**Evaluation** For fair comparison, we used a three-shot setting in all comparison experiments. For natural language proofs, correctness was manually verified by experts. For formal proofs in Lean4, correctness was automatically verified. We report the number of problems solved and the corresponding success rate for each dataset.

### 4.2 Main Results

Our experiments reveal a notable pattern: the integration of symbolic reasoning and neural approaches creates a system far more powerful than either approach alone. As shown in Table 1, while state-of-the-art language models struggle with the precision required for inequality proofs, and symbolic methods alone lack creative problem-solving abilities, IneqSearch combines the strengths of both worlds.

Analysis shows that the most powerful reasoning language model (o3-mini) solves only 35% and 15% of problems in INEQ-437 and MO-INT-20, respectfully, highlighting the limitations of purely neural approaches. Notably, language models demonstrate substantial difficulty in producing formal Lean4 proofs, with success rates below 1%. In contrast, IneqSearch achieves a 78.3% success rate on these challenging inequalities, demonstrating the effectiveness of the integrated approach in mathematical reasoning tasks. Hybrid methods like AIPS and LIPS show improved performance over pure LLM

Table 2: Ablation study on IneqSearch components.

| ECPD | SOS | Target Shift | Exploration | Iterative Learning | Problems Solved |
|:---:|:---:|:---:|:---:|:---:|:---:|
| ✓ | – | – | – | – | 126 |
| ✓ | ✓ | – | – | – | 257 |
| ✓ | ✓ | ✓ | – | – | 292 |
| ✓ | ✓ | – | ✓ | – | 325 |
| ✓ | ✓ | ✓ | ✓ | – | 334 |
| ✓ | ✓ | ✓ | ✓ | ✓ | 342 |

approaches but still fall short of IneqSearch's capabilities. By integrating inductive exploration with rigorous deduction, IneqSearch ensures both correctness and interpretability in its proofs. Detailed error analysis of LLM-generated proofs is provided in Appendix E.

**Performance on High-Difficulty Problems**   IneqSearch shows strong performance on problems that are difficult for existing methods. On the MO-INT-20 dataset, which contains real inequalities from Olympiad competitions, many problems require multi-step scaling and are often non-symmetric and non-homogeneous (see Appendix D). These characteristics pose significant challenges for symbolic solvers and LLMs.

First, symbolic solvers often failed to directly solve such problems. For example, the Polish-2004 problem requires the use of the Hölder inequality, while USAMO 2017 P6 involves the tangent line trick. Purely algebraic techniques, such as direct decomposition, rarely uncover these methods. Without such approaches, analyze the original expressions becomes extremely difficult.

Second, some problems require substantial algebraic manipulation and comparison. For instance, in IMO 2020 P2, even after applying the AM-GM inequality, the proof involves adding, subtracting, and comparing over thirty monomials. LLMs can often identify where AM-GM should be applied, but they usually cannot complete the subsequent deductive steps. In some cases, the models even doubts the correctness of its initial step.

These challenges persist even for state-of-the-art hybrid approaches. While other methods such as AIPS and LIPS are only able to solve a subset of the dataset, IneqSearch, in contrast, achieves full coverage by successfully proving all 20 problems.

## 4.3   Ablation Study

Table 2 presents a systematic ablation analysis of IneqSearch's components, highlighting their individual contributions to the system's performance. The baseline model, which uses only the symbolic solver, solves 257 problems. The integration of target shift for homogenization provides a marginal improvement, solving 35 additional problems. The inclusion of LLM exploration further boosts performance, increasing the number of solved problems to 334. Finally, the iterative learning mechanism resolves 8 additional complex problems that were previously unsolved, achieving a total of 342 problems solved. These results demonstrate the complementary nature of the system's components and the effectiveness of integrating symbolic and neural approaches.

## 4.4   Iterative Progress Analysis

The performance of IneqSearch demonstrates consistent improvement across multiple learning iterations. Figure 4 quantifies this progression, documenting an increase from 257 successfully solved problems initially to 342 problems after multiple rounds of theorem acquisition and proof refinement. In each round, IneqSearch traverses the entire benchmark and attempts to prove each inequality within it. The theorem base will be updated once after completing the round, ensuring that any newly discovered results become available as tools for subsequent iterations.

This iterative improvement follows a pattern consistent with incremental knowledge acquisition, where each round builds upon previously established results. By systematically incorporating new theorems into its proof, IneqSearch is able to unlock solutions to problems that were previously inaccessible. Figure 5 illustrates a representative case of this process, where theorem $TH_{82}$, estab-

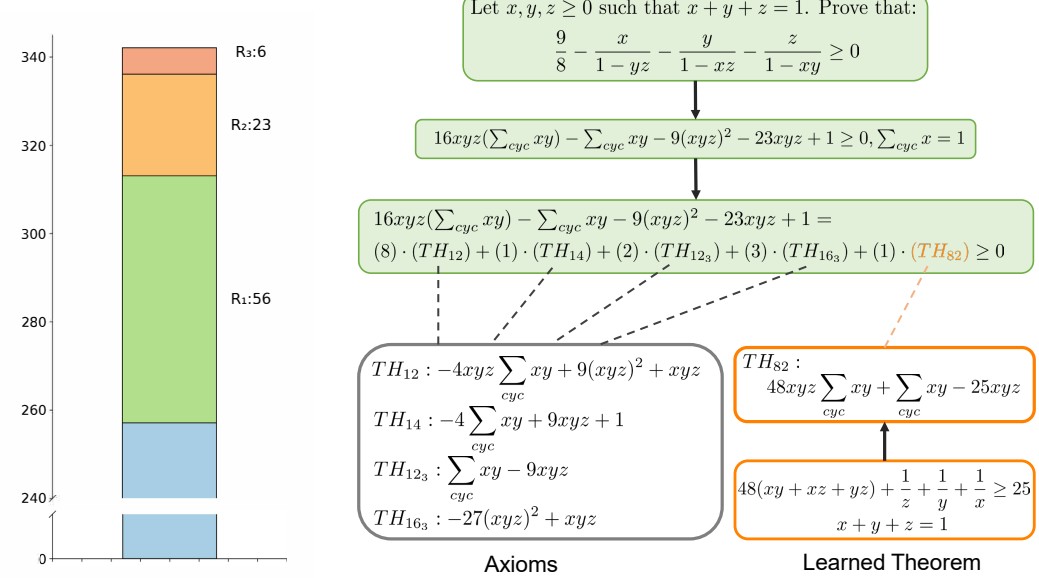

Figure 4: Visualization of cumulative problems solved across four rounds.

Figure 5: IneqSearch successfully proves a hard problem leveraging a previous proven inequality.

lished in earlier rounds, enables the solution of a previously unresolved inequality, exemplifying the practical benefit of cumulative learning.

The observed improvement pattern demonstrates the effectiveness of the system's knowledge accumulation mechanism, where each new theorem potentially extends the range of solvable problems. The continual expansion of mathematical tools underlies the robust performance gains observed throughout the iterative process.

## 4.5 Computation Cost

IneqSearch first attempts to prove each inequality using the symbolic solver. If this attempt fails, the system proceeds to call the LLM for further exploration. In our implementation, there are 5 inequality lemmas to be explored, and each inequality exploration calls the LLM once, with one additional retry if the first call is invalid. The exploration of inequality lemmas is conducted in parallel, allowing us to collect all possible valid scaled expressions after exploration.

Table 3 summarizes the average runtime and time budget for the proving steps of IneqSearch with o3-mini on our INEQ-437 benchmark. Only the major time-consuming steps are listed, and the steps are not strictly sequential in the implementation. Other routine algebraic manipulations are omitted, as their runtimes are comparatively negligible. The reported runtime for each process refers to the average time per single execution, although each process may be called multiple times within a proof. For accurate measurement of actual solving time, we report the runtime for successful proofs, where IneqSearch requires only 12.3 seconds on average per problem, demonstrating **high efficiency**.

Table 4 presents the average solving time for successfully solved problems and compares our method to LIPS and o3-mini. Compared to LIPS, our approach achieves more than a $100\times$ speedup in runtime. While these results are convincing in practice, we note that this direct runtime comparison is not entirely fair from an algorithmic perspective, as the majority of the time cost in LIPS arises from repeated LLM calls. Our method is primarily solver-driven and typically requires far fewer LLM calls per problem.

Table 3: Runtime analysis (in seconds) of the proof workflow on INEQ-437.

| Module | Process | Avg. Runtime | Time Budget |
|---|---|---|---|
| Solver | ECPD Transform | 0.6 | 30 |
| | Theorem Search | 41.6 | 240 |
| | SOS | 0.4 | 30 |
| | Total | 12.3 | 300 |
| Inductive Agent (o3-mini) | Equality Condition | 17.3 | 180 |
| | Target Shift | 19.1 | 180 |
| | AM-GM Ineq. Expl. | 32.6 | 180 |
| | Cauchy-Schwarz Ineq. Expl. | 27.3 | 180 |
| | Hölder's Ineq. Expl. | 41.0 | 180 |
| | Jensen's Ineq. Expl. | 34.4 | 180 |
| | Tangent Line Ineq. Expl. | 37.9 | 180 |
| | Total | 65.8 | 300 |
| Overall | | 24.1 | 900 |

Table 4: Comparison of solving time (in seconds) and success rate.

| Methods | Avg. Runtime | Success Rate |
|---|---|---|
| OpenAI o3-mini | 63 | 35.2% |
| LIPS w/ 120 min budget | 5187 | 62.5% |
| LIPS w/ 90 min budget | 2946 | 50.3% |
| IneqSearch w/o LLM | 12 | 66.8% |
| IneqSearch w/ o3-mini | 24 | 78.3% |

## 4.6 Error Case in LLM's Output

Our analysis of LLM-generated proofs reveals critical consistency issues, underscoring the necessity of formal verification in mathematical reasoning. When proving complex inequalities, LLMs are prone to making subtle yet critical errors in logical deductions or algebraic calculations. For example, in solving the inequality $\sqrt{6x+1} + \sqrt{6y+1} + \sqrt{6z+1} \le 3\sqrt{3}$, Gemini2 produces contradictory results in two attempts at calculating derivatives, as shown in Figure 6.

Such inconsistencies suggest that while LLMs can mimic advanced mathematical reasoning, they require an automated verifier to ensure the correctness of steps and prevent hallucinations, especially when solving problems that require rigorous deductive reasoning. Therefore, integrating rigorous verification mechanisms is essential for both the reliability and trustworthiness of LLM-assisted mathematical problem solving.

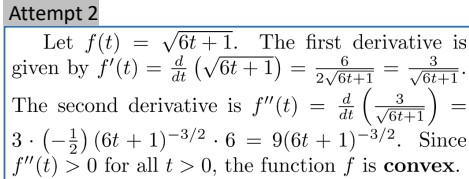

Figure 6: LLM generates inconsistent content in two attempts.

## 5 Discussion

IneqSearch implements a critical principle for automated mathematical reasoning: continuously acquiring and applying mathematical knowledge. Its improved performance shows that effective reasoning relies on leveraging established results to construct new proofs, reflecting the cumulative nature of mathematics.

A promising research direction involves utilizing LLMs to extract generalizable patterns from specific problems. This approach aligns with established mathematical methodology: observation of patterns, formulation of conjectures, and subsequent verification. For example, consider the following IMO problem: for positive real numbers $a, b, c, d$ such that $ab + bc + cd + da = 1$, show that $\frac{a^3}{b+c+d} + \frac{b^3}{c+d+a} + \frac{c^3}{d+a+b} + \frac{d^3}{a+b+c} \geq \frac{1}{3}$. This problem can be extended to a broader class of inequalities, such as $\sum_i^n \frac{x_i^3}{s - x_i} \geq \frac{4k^2}{(n-1)n}$ where $s = \sum_i x_i$ and $k^2 = \sum_{i<j} x_i x_j$. The key insight is recognizing that denominators of the form $s - x_i$ and the cyclic condition $ab + bc + cd + da = 1$ can be abstracted to the setting $k^2 = \sum_{i<j} x_i x_j$, representing an elementary symmetric polynomial of degree 2. Maintaining homogeneity during this abstraction process is essential. Such generalizations require pattern recognition and analytical reasoning capabilities well-suited to LLMs' strengths.

# 6 Related Work

Our work integrates advances in mathematical reasoning, automated theorem proving, and Olympiad problem solving.

## 6.1 Mathematical Reasoning

Automated mathematical reasoning has progressed through several complementary approaches. Benchmarks like MATH [20] and GSM8K [21] have established standards for evaluating reasoning capabilities. Advances in deep learning for symbolic mathematics [22] and automated theorem proving [23, 24, 25, 17] have emerged through neural-guided proof assistants. Formal systems like Lean [26, 27, 28] provide verification frameworks with extensive theorem libraries. Additional approaches include curriculum learning [29] and step-by-step verification [6]. Despite these advances, inequality proving remains challenging due to its requirements for precise algebraic manipulation and problem-specific insights.

## 6.2 Olympiad Problem Solving

Recent innovations in automated Olympiad problem-solving include formal mathematics benchmarks like MiniF2F [7] and multimodal challenges in OlympiadBench [2]. Domain-specific systems such as AlphaGeometry [3] have demonstrated success in plane geometry through learned construction of auxiliary elements. For inequality problems, LIPS [19] combines language models for expression rewriting with symbolic optimization tools, while AIPS [1] explores autonomous theorem generation. Our work extends these approaches by implementing a systematic framework for inequality decomposition with iterative theorem acquisition, creating a self-improving system for mathematical reasoning.

# 7 Conclusion and Future Work

This work offers three main contributions: (1) it reformulates inequality proving as a theorem-based decomposition search, enabling the systematic breakdown of complex inequalities into manageable subproblems using established mathematical results; (2) it develops a hybrid system that combines symbolic verification with LLM-guided exploration, leveraging both rigorous algorithmic reasoning and the flexible problem-solving capabilities of LLMs; and (3) it implements an iterative learning mechanism that continuously improves the system through ongoing theorem acquisition, allowing it to expand its knowledge base and enhance its proving abilities over time. The effectiveness of this approach is demonstrated by the system's strong performance on challenging Olympiad inequalities, including complete coverage of the MO-INT-20 dataset.

Future work should extend this methodology to other mathematical domains and further enhance generalization capabilities. The framework of decomposing complex problems, searching for established patterns, and incorporating new knowledge represents a generalizable approach to mathematical reasoning, with applications in formal verification and automated problem solving.

## Acknowledgments

The research reported in this paper was partially supported by the National Social Science Foundation of China (20&ZD047).

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

# Appendix

## A Proof Methodologies

### A.1 Elementary Cyclic Polynomial Decomposition

Elementary Cyclic Polynomial Decomposition (ECPD) is a calculation-based approach designed for proving cyclic inequalities. This method enables us to leverage computational tools and to build a unified-format theorem library for addressing complex algebraic inequalities, extending beyond cyclic cases alone.

A fundamental principle in algebraic inequality theory is that the structure of the polynomial—whether symmetric or merely cyclic—determines the most suitable basis for its decomposition and analysis.

**Theorem 1.** *Let $T(x_1, x_2, \ldots, x_n)$ be a symmetric polynomial in $n$ variables. Then $T$ can always be expressed as a polynomial in terms of the elementary symmetric polynomials $S_1, S_2, \ldots, S_n$, where*

$$S_1 = \sum_{i=1}^{n} x_i, \quad S_2 = \sum_{1 \leq i < j \leq n} x_i x_j, \quad \ldots, \quad S_n = x_1 x_2 \ldots x_n.$$

This classical result allows for the reduction of symmetric inequalities to expressions involving only the elementary symmetric polynomials. Such a reduction significantly simplifies both theoretical analysis and computational verification, as the search space for potential equalities or extreme cases is greatly narrowed.

For polynomials exhibiting cyclic symmetry, a parallel decomposition exists, which forms the backbone of the ECPD method:

**Theorem 2.** *Let $T(x_1, x_2, \ldots, x_n)$ be a cyclic polynomial of in $n$ variables. Then $T$ can always be expressed as a polynomial in terms of the elementary cyclic polynomials $C_{k_1, k_2, \ldots, k_m}^{(n)}$, where*

$$C_{k_1, k_2, \ldots, k_m}^{(n)} = \sum_{cyc} x_1^{k_1} x_2^{k_2} \ldots x_m^{k_m},$$

*with $m \leq n$ and $k_1 + k_2 + \ldots + k_m = \deg T$.*

The introduction of the elementary cyclic basis enables systematic treatment of cyclic inequalities, especially when the expressions are not fully symmetric. By expressing any cyclic polynomial in this basis, we unlock the potential for automated computation and algorithmic proof strategies, as well as a unified language for theorems and results.

### A.2 Sum of Squares

One of the fundamental procedures for proving inequalities is to rewrite them as a sum of squares, thereby establishing the non-negativity of a given expression. This property forms the foundation of the SOS method.

**Theorem 3** (Hilbert). *Let $T(x_1, x_2, \ldots, x_n)$ be a polynomial.*

- *If $T \geq 0$ and is homogeneous, then $T$ can be expressed as*

$$T = \sum_k t_k^2,$$

  *where $t_k$ are certain real rational functions.*

- *If $T \geq 0$ and is non-homogeneous, and $\{x_i\}_{i=1}^{n}$ satisfies the condition $\sum x_i \leq 1$, then $T$ can be expressed as*

$$T = \sum_k c_k \, x_1^{\alpha_1} x_2^{\alpha_2} \ldots x_n^{\alpha_n} \left(1 - \sum_i x_i\right)^{\alpha_{n+1}},$$

  *where $\{\alpha_i\}_{i=1}^{n+1}$ are non-negative integers and $c_k > 0$.*

These theorems are qualitative results that help establish the existence of a method to prove that a polynomial is non-negative. For homogeneous functions, this is achieved by analyzing sums of squares, while for non-homogeneous functions, it involves expressing the polynomial as a sum of non-negative quantities. In [30], this principle is implemented based on an explicit semidefinite program.

**SDPSOS**  Given a real polynomial $T(x_1, \ldots, x_n)$, together with inequality constraints $g_1(x) \geq 0, \ldots, g_k(x) \geq 0$ and equality constraints $h_1(x) = 0, \ldots, h_l(x) = 0$, the SDPSOS method seeks to certify nonnegativity of $T$ under these constraints by constructing a SOS representation:

$$T(x) = \sum_{i=0}^{k} s_i(x)\, g_i(x) + \sum_{j=1}^{l} t_j(x)\, h_j(x),$$

where $g_0(x) := 1$, each $s_i(x)$ is an SOS polynomial, and $t_j(x)$ are arbitrary polynomials. Each SOS term can be written as $s_i(x) = v_i(x)^\top M_i v_i(x)$, where $M_i$ is a symmetric positive semidefinite matrix and $v_i(x)$ is the vector of monomial basis elements up to a prescribed degree.

Thus, certifying nonnegativity reduces to finding matrices $M_0, M_1, \ldots, M_k \succeq 0$ and polynomials $t_j(x)$ such that

$$T(x) = \sum_{i=0}^{k} v_i(x)^\top M_i v_i(x)\, g_i(x) + \sum_{j=1}^{l} t_j(x)\, h_j(x),$$

with all coefficients matching when both sides are expanded. The entries of the Gram matrices $M_i$ and the coefficients of $t_j(x)$ become variables in a system of linear equations, while the positive semidefiniteness of $M_i$ is imposed as a semidefinite constraint.

Therefore, the problem of certifying nonnegativity via SOS reduces to the following explicit semidefinite program (SDP):

*Find symmetric matrices $M_0, M_1, \ldots, M_k$ and polynomials $t_1(x), \ldots, t_l(x)$*
*such that*

$$T(x) = \sum_{i=0}^{k} v_i(x)^\top M_i v_i(x)\, g_i(x) + \sum_{j=1}^{l} t_j(x)\, h_j(x)$$

*subject to $M_i \succeq 0, \; i = 0, \ldots, k.$*

This modeling is possible because the set of all positive semidefinite matrices forms a convex cone, and the set of solutions to the linear equations is an affine subspace; their intersection is a convex set. Therefore, the problem becomes a convex optimization problem with linear equality constraints and matrix positivity constraints. If $T(x)$ and the constraints exhibit symmetry, equivalent terms under the symmetry group can be identified, reducing the size of the SDP. Additionally, if $T(x)$ vanishes at certain points (equality cases), the corresponding Gram matrices are further constrained to have these directions in their kernel, reflecting additional algebraic structure.

### A.3   Axioms in Theorem Database

IneqSearch is designed to evolve from basic problem-solving capabilities to an advanced level of expertise through continuous learning and adaptation. Initially, the theorem database contains a set of fundamental theorems from Murihead's inequality, Shur's inequality and Vasc's inequality, which serve as axioms.

**Theorem 4** (Muirhead's Inequality). *Let $x_1, x_2, \ldots, x_n$ be non-negative real numbers, and let $\mathbf{a} = (a_1, \ldots, a_n)$ and $\mathbf{b} = (b_1, \ldots, b_n)$ be two sequences of non-negative integers in non-increasing order. If $\mathbf{a}$ majorizes $\mathbf{b}$, that is,*

$$\sum_{i=1}^{k} a_i \geq \sum_{i=1}^{k} b_i \quad \text{for all } 1 \leq k < n, \qquad \text{and} \qquad \sum_{i=1}^{n} a_i = \sum_{i=1}^{n} b_i,$$

*then the following inequality holds:*

$$\sum_{\sigma \in S_n} x_1^{a_{\sigma(1)}} x_2^{a_{\sigma(2)}} \ldots x_n^{a_{\sigma(n)}} \geq \sum_{\sigma \in S_n} x_1^{b_{\sigma(1)}} x_2^{b_{\sigma(2)}} \ldots x_n^{b_{\sigma(n)}},$$

*where $S_n$ denotes the set of all permutations on $n$ elements.*

In this work, we develop an automated pipeline for the systematic generation of Muirhead-type inequalities. Given the number of variables $n$ and the total degree $d$, the algorithm enumerates all possible integer partitions $(a_1, a_2, \ldots, a_n)$ in non-increasing order with $a_1 + \ldots + a_n = d$. For each pair of partitions $(\mathbf{a}, \mathbf{b})$ such that $\mathbf{a}$ majorizes $\mathbf{b}$, the corresponding difference of symmetric sums is constructed:

$$F_{\mathbf{a},\mathbf{b}}(x_1, \ldots, x_n) = \sum_{\sigma \in S_n} x_1^{a_{\sigma(1)}} \ldots x_n^{a_{\sigma(n)}} - \sum_{\sigma \in S_n} x_1^{b_{\sigma(1)}} \ldots x_n^{b_{\sigma(n)}}.$$

Each inequality is then canonicalized by factoring and removing any constant multipliers. To ensure minimality, the collection is further filtered to eliminate redundant inequalities via numerical evaluation and linear dependence checks.

## B Inductive Agent Configuration

### B.1 Prompt Template

Here, we present the prompt template used to automate the application of the Cauchy-Schwarz inequality with the LLM. Other exploration prompts are similar and we will include them at the revised version.

---

**Prompt for Cauchy-Schwarz Inequality Exploration**

We are solving an inequality problem. Please do not use the code interpreter.
The inequality is: {S_str}
We have the following important information:
1. Equality holds for: {cond_str}
2. Assumption condition: {assump_str}

Now, we want to use the Cauchy-Schwarz inequality to scale $S$, so that we can obtain an expression which is easier to handle. The term indices are: {term_dict}
Recall that the Cauchy-Schwarz inequality is:
$(\sum_{i=1}^{n} A_i)(\sum_{i=1}^{n} B_i) \geq (\sum_{i=1}^{n} C_i)^2$ or $\sum_{i=1}^{n} C_i \leq \sqrt{(\sum_{i=1}^{n} A_i)(\sum_{i=1}^{n} B_i)}$, where $C_i = \sqrt{A_i B_i}$

Follow these steps to determine whether the original inequality is suitable for applying the Cauchy-Schwarz inequality:
1. Identify the terms that are suitable for the Cauchy-Schwarz inequality, and return their term indices.
2. Determine the role of these terms as $A_i$ or $C_i$ based on their coefficients: $A_i$ for positive, $C_i$ for negative.
(1) If the role is $A_i$, use the first form of the given Cauchy-Schwarz inequality; otherwise, use the second.
(2) Ensure that the role ($A_i$ or $C_i$) is included among the terms of $S$.
3. Design $B_i$ so that the other role ($A_i$ or $C_i$) is easy to handle.
4. If the Cauchy-Schwarz inequality is applicable, return the corresponding term indices as a list of integers.

Provide your analysis and calculation steps, and summarize your answer in a JSON. The JSON format begins with ```json as shown below. Each term should be a LaTeX string without the $ symbol, and the product sign * should be explicitly written between parentheses in the expressions:
```json
{
"applicable": true or false,
"role": "A_i" or "C_i",
"term_numbers": [num1, num2, ...] corresponding to the indices of $A_1/C_1$, $A_2/C_2$, ...
"B_1": latex formula corresponding to $A_1$ or $C_1$,
"B_2": latex formula corresponding to $A_2$ or $C_2$,
...
}
```

---

For each input inequality, we decompose it into individual algebraic terms, assign each term a unique index, and represent them in LaTeX format as a dictionary (`term_dict`). For example, for the inequality $a^2 + b^2 + c^2 \geq ab + bc + ca$, the `term_dict` would be {1: "a^2", 2: "b^2", 3: "c^2", 4: "-ab", 5: "-bc", 6: "-ca"}. We also provide the equality condition (`cond_str`) and the assumptions (`assump_str`) as LaTeX strings in the prompt, ensuring the model has access to all

relevant information during reasoning. The prompt instructs the model to analyze the structure of each term to determine its role in the Cauchy-Schwarz inequality as either $A_i$ or $C_i$, and to systematically design the corresponding $B_i$ for each.

## B.2 Post-processing

In the model response, we extract the relevant terms from `term_dict` according to the `term_numbers` in the JSON, and combine them with the extracted $B_i$ expressions to automatically assemble a standard Cauchy-Schwarz inequality. For example, if the role is $A_i$, we construct

$$\left(\sum_i A_i\right)\left(\sum_i B_i\right) \geq \left(\sum_i \sqrt{A_i B_i}\right)^2 .$$

Any error encountered during this construction process (as detected by our multiple symbolic checks) is regarded as an exploration failure.

## C  Limitation

Theorem search method equipped with ECPD and SOS is theoretically capable of solving all inequality problems within Equation 1. However, in practice, the search process is often challenging. Our methodology relies on the decomposition identity $F = \sum F_i$, where a necessary condition for a successful search is that the equality conditions must be satisfied for all subcomponents:

Suppose $\mathbf{x} = (x_1, x_2, \ldots, x_n)$ such that $F(\mathbf{x}) = 0$.

Then for every $i$, from $F_i(\mathbf{x}) \geq 0$ , we have $F_i(\mathbf{x}) = 0$.

For many hard inequalities, the equality conditions are uncommon or may not exist in $\mathbb{R}^+$, necessitating case-by-case analysis. For example, consider proving the inequality $12(\sum_{cyc} x^2 y^2)(\sum_{cyc} x^3) \leq \sum_{cyc} xy$ under the assumption $x + y + z = 1$. The equality holds if and only if $(x, y, z) = (0, 0, 1)$ up to permutation. Then all theorems $g$ for which $g(0, 0, 1) \neq 0$ will not be applicable.

Such cases demonstrate that equality conditions can be highly restrictive. Consequently, if there are not enough applicable theorems with this equality condition in the database, it will limit the applicability and effectiveness of the search in practice.

## D  Additional Examples

Here are some examples drawn from MO-INT-20 dataset [1] for our method illustration.

- **IMO 1990 Shortlist**
  For $a > 0$, $b > 0$, $c > 0$, $d > 0$ such that $ab + bc + cd + da = 1$, show that:
  $$\frac{a^3}{b+c+d} + \frac{b^3}{c+d+a} + \frac{c^3}{d+a+b} + \frac{d^3}{a+b+c} \geq \frac{1}{3}$$

- **Poland 2004:**
  For $a > 0$, $b > 0$, $c > 0$, $d > 0$, show that:
  $$\frac{a}{(a^3 + 63bcd)^{\frac{1}{3}}} + \frac{b}{(63acd + b^3)^{\frac{1}{3}}} + \frac{c}{(63abd + c^3)^{\frac{1}{3}}} + \frac{d}{(63abc + d^3)^{\frac{1}{3}}} \geq 1$$

- **USAMO 2017 P6:**
  For $a \geq 0$, $b \geq 0$, $c \geq 0$, $d \geq 0$ such that $a + b + c + d = 4$, show that:
  $$\frac{a}{b^3 + 4} + \frac{b}{c^3 + 4} + \frac{c}{d^3 + 4} + \frac{d}{a^3 + 4} \geq \frac{2}{3}$$

- **IMO 2020 P2:**
  For $a \geq b$, $b \geq c$, $c \geq d$, $d > 0$ such that $a + b + c + d = 1$, show that:
  $$(a + 2b + 3c + 4d)a^a b^b c^c d^d < 1$$

# E  LLM Proof Generation

For the Lean4 code generation task, each problem is initially translated into Lean4 code using a fixed template. The LLMs are then prompted to complete the code. One example Lean4 code is:

```
import Mathlib

theorem inequality_problem
(x y z : ℝ)
(hx : x > 0) (hy : y > 0) (hz : z > 0) (cond : x +
    y + z = 1):
(1 + 1 / x) * (1 + 1 / y) * (1 + 1 / z) ≥ 64:= by
```

The corresponding response of o1-mini is provided below.

```
import Mathlib

theorem inequality_problem
  (x y z : ℝ)
  (hx : x > 0) (hy : y > 0) (hz : z > 0)
  (cond : x + y + z = 1) :
  (1 + 1/x) * (1 + 1/y) * (1 + 1/z) ≥ 64:= by
  have h_sym : x = y ∧ y = z :=
    by
      let f := λ (x y z : ℝ)
      (1 + 1/x) * (1 + 1/y) * (1 + 1/z)
      sorry
  cases h_sym with hxy hyz
  subst hxy
  subst hyz
  have h_val : x = 1/3 := by
    rw [←cond] at hyz
    linarith
  rw h_val
  simp
  norm_num
  exact le_refl 64
```

Based on the messages from the Lean4 server, the code contains several errors. These errors include an *Invalid Pattern* caused by the use of an improper constructor, *Unsolved Goals* due to the lack of a proof for the symmetry assumption ($x = y \land y = z$), and an *Unexpected Token* error arising from the use of $*$ instead of a valid logical operator. These issues indicate issues with pattern matching, incomplete proofs, and syntax within the code.

Our analysis of the outputs for several problems reveals frequent errors, such as confusion between Lean3 and Lean4 syntax, including the misuse of `begin` and `Real.sq_nonneg`. For more complex problems, the models often fail to perform even basic logical reasoning during code generation.

In the natural language testing setup, problems are presented in LaTeX format, and LLMs are tasked directly with generating proofs. The proof produced by o1-mini is presented in Figure 7, which successfully solves the problem.

To prove that for positive real numbers $x, y, z$ satisfying $x + y + z = 1$, the inequality

$$\left(1 + \frac{1}{x}\right)\left(1 + \frac{1}{y}\right)\left(1 + \frac{1}{z}\right) \geq 64$$

holds, we use **Jensen's Inequality** with a convex function.

## Key Steps

1. Rewrite the product as:

$$\left(1 + \frac{1}{x}\right)\left(1 + \frac{1}{y}\right)\left(1 + \frac{1}{z}\right) = \left(\frac{x+1}{x}\right)\left(\frac{y+1}{y}\right)\left(\frac{z+1}{z}\right)$$

2. Take natural logarithms:

$$\ln\left[\left(1 + \frac{1}{x}\right)\left(1 + \frac{1}{y}\right)\left(1 + \frac{1}{z}\right)\right] = \ln\left(1 + \frac{1}{x}\right) + \ln\left(1 + \frac{1}{y}\right) + \ln\left(1 + \frac{1}{z}\right)$$

3. Use the convex function $f(t) = \ln\left(1 + \frac{1}{t}\right)$, whose second derivative $f''(t) = \frac{1}{[t(t+1)]^2} > 0$, to apply **Jensen's Inequality**:

$$\frac{\ln\left(1 + \frac{1}{x}\right) + \ln\left(1 + \frac{1}{y}\right) + \ln\left(1 + \frac{1}{z}\right)}{3} \geq \ln\left(1 + \frac{3}{1}\right) = \ln(4)$$

4. Multiply by 3 and exponentiate:

$$\ln\left[\left(1 + \frac{1}{x}\right)\left(1 + \frac{1}{y}\right)\left(1 + \frac{1}{z}\right)\right] \geq \ln(64) \quad \Rightarrow \quad \left(1 + \frac{1}{x}\right)\left(1 + \frac{1}{y}\right)\left(1 + \frac{1}{z}\right) \geq 64$$

## Conclusion

Equality holds when $x = y = z = \frac{1}{3}$, satisfying $x + y + z = 1$. Thus:

$$\left(1 + \frac{1}{x}\right)\left(1 + \frac{1}{y}\right)\left(1 + \frac{1}{z}\right) \geq 64$$

for all $x, y, z > 0$ with $x + y + z = 1$.

Figure 7: Natural language proof of o1-mini. It correctly adopts Jensen's Inequality and successfully solves the problem.

