# OpenReview forum: "IneqSearch: Hybrid Reasoning for Olympiad Inequality Proofs"
_NeurIPS.cc/2025/Conference — NeurIPS 2025 poster_

### Official Review · Reviewer_Hdro · 2025-06-24

**Clarity:** 2
**Significance:** 3
**Originality:** 3
**Rating:** 4
**Confidence:** 4

**Summary:**

This paper presents IneqSearch, an inequality prover that utilizes decomposition techniques. Specifically, IneqSearch employs a symbolic solver to identify which theorems are involved in the proof and leverages Large Language Models (LLMs) to instantiate and establish a set of candidate theorems. Additionally, an iterative learning mechanism can be seamlessly integrated into the framework, allowing the theorem library to evolve automatically. Experimental results demonstrate that the proposed method achieves a significant improvement on two Olympiad-level benchmarks.

**Questions:**

- The authors mention that the natural language solutions generated by LLMs are manually checked by experts. What are the criteria used for evaluating these solutions?
- The SOS and theorem searcher incorporate numerical optimization methods for semi-definite programming and linear programming. How is the discrepancy between numerical results and symbolic results addressed? For instance, if $λ_i$ is determined to be $1.414$, will it be converted to $\sqrt{2}$?
- Figure 1 lacks clarity; it would be beneficial to enhance the illustration of the workflow.
- More experimental details are needed. For example, what is the average solving time? What prompts were used for generating natural language solutions? Additionally, will the code be made open-source in the end?

**Ethical Concerns:**

["NO or VERY MINOR ethics concerns only"]

**Final Justification:**

I lean toward borderline acceptance. (1) The proposed symbolic method and experimental results are promising. (2) While I appreciate the self-improvement concept, the current implementation is too limited to justify a higher score.

**Limitations:**

Yes

**Quality:**

3

**Strengths And Weaknesses:**

### Strengths
- The decomposition-based framework, along with a neuro-symbolic methodology for inequality proving, shows great promise.
- The framework incorporates an iterative learning strategy, offering an intriguing approach for the self-evolution of the neuro-symbolic system.
- It demonstrates strong performance on competition-level benchmarks, indicating a promising direction for automatic inequality proving.

### Weaknesses
- Although the proposed method is extremely effective, it appears to be limited to the domain of inequality proving. It would be beneficial to discuss how this method could inspire advancements in other related fields, such as general mathematical reasoning or mathematical theorem proving.
- The decomposition-based proving approach, particularly the SOS technique, generates proofs that lack human readability and explicitness. This limitation hinders the method's effectiveness as an educational tool for providing answers and affects its performance when used as training data to enhance models.
- Evaluating the method on additional benchmarks could further demonstrate its effectiveness. For example, the 567Neq benchmark introduced in LIPS would be a valuable test case.
- The ablation study indicates that the current iterative learning mechanism has limitations, as it only successfully solves eight additional problems when integrated into the method.

---

> ### Author Rebuttal · Authors · 2025-07-31
>
> Thanks for your review.
>
> Even though DeepMind recently used Gemini (with natural language) to win a gold medal in the IMO, our results show that LLMs still face significant limitations in Olympiad inequality proofs. Algebraic inequality proving is exceptionally challenging because it involves navigating an infinite reasoning space and requires both rigorous deductive reasoning and constructive proof techniques. Combining symbolic computation with LLMs enables two complementary modes of thought—rigorous deduction and creative exploration—making this hybrid approach especially powerful for tackling inequalities.
>
> # Takeaways
>
> 1. Relying solely on LLMs to generate natural or formal proofs remains challenging. Our work demonstrates that integrating symbolic solvers with LLMs enables hybrid reasoning, resulting in substantial performance improvements. T
>
> 2. The decomposition and search framework offers a promising direction for automated inequality proving. By leveraging LLMs for implementation within this framework, our results establish a practical path forward for general automated inequality proving.
>
> # Question response
> ## Question 1
>
> For the evaluation, no step credit is given. Only fully correct or incorrect responses are considered. In fact, errors in LLM-generated natural language proofs are often methodological such as employing incorrect logic, reversing the direction of inequalities, or using specific examples in place of general proofs, rather than minor technical mistakes.
>
> ## Question 2
> For linear programming, the variable domain is set to rational. For SDP, numerical values are rationalized using SymPy's `nsimplify` function with a tolerance of 1e-15.
> It should be noted that decomposition equalities are checked using symbolic computation to ensure soundness.
>
> ## Question 3
>
> Please specify which parts you find confusing, so that we can improve our paper.
>
>
> ## Question 4
> #### Computation time
> **Table 3.** Runtime analysis of the proof workflow on INEQ-437.
> | Module                    | Process                      | Avg. Runtime (s) | Time Budget (s) |
> |---------------------------|------------------------------|------------------|-----------------|
> |                           | ECPD Transform               | 0.6              | 30              |
> |                           | Theorem Search               | 41.6             | 240             |
> |                           | SOS                          | 0.4              | 30              |
> | Solver                    | **Total**                    | **12.3**         | **300**         |
> |                           |                              |                  |                 |
> |                           | Equality Condition           | 17.3             | 180             |
> |                           | Target Shift                 | 19.1             | 180             |
> |                           | AM-GM Ineq. Expl.            | 32.6             | 180             |
> |                           | Cauchy-Schwarz Ineq. Expl.   | 27.3             | 180             |
> |                           | Hölder's Ineq. Expl.         | 41.0             | 180             |
> |                           | Jensen's Ineq. Expl.         | 34.4             | 180             |
> |                           | Tangent Line Ineq. Expl.     | 37.9             | 180             |
> | Inductive Agent (o3-mini) | **Total**                    | **65.8**         | **300**         |
> |                           |                              |                  |                 |
> | **Overall**               |                              | **24.1**         | **900**         |
>
> ---
>
> **Table 4.** Comparison of solving time and success rate.
> | Methods                  | Avg. Runtime (s) | Success Rate |
> |--------------------------|------------------|--------------|
> | OpenAI o3-mini           | 63               | 35.2%        |
> | LIPS w/ 120 min budget   | 5187             | 62.5%        |
> | LIPS w/ 90 min budget    | 2946             | 50.3%        |
> | IneqSearch w/o LLM       | 12               | 66.8%        |
> | IneqSearch w/ o3-mini    | 24               | 78.3%        |
>
>
> ---
>
> IneqSearch first attempts to prove each inequality using the symbolic solver. If this attempt fails, the system proceeds to call the LLM for further exploration. In our implementation, there are 5 inequality lemmas to be explored, and each inequality exploration calls the LLM once, with one additional retry if the first call is invalid. The exploration of inequality lemmas is conducted in parallel, allowing us to collect all possible valid scaled expressions after exploration.
>
> Table 3 summarizes the average runtime and time budget for the proving steps of IneqSearch with o3-mini on our INEQ-437 benchmark. Only the major time-consuming steps are listed, and the steps are not strictly sequential in the implementation. Other routine algebraic manipulations are omitted, as their runtimes are comparatively negligible. The reported runtime for each process refers to the average time per single execution, although each process may be called multiple times within a proof. For accurate measurement of actual solving time, we report the runtime for successful proofs, where IneqSearch requires only 12.3 seconds on average per problem, demonstrating **high efficiency**.
>
> Table 4 presents the average solving time for successfully solved problems and compares our method to LIPS and o3-mini. Compared to LIPS, our approach achieves more than a 100× speedup in runtime. While these results are convincing in practice, we note that this direct runtime comparison is not entirely fair from an algorithmic perspective, as the majority of the time cost in LIPS arises from repeated LLM calls. Our method is primarily solver-driven and typically requires far fewer LLM calls per problem.
>
> Regarding computational complexity, IneqSearch relies on several computational tools, and the precise complexity depends on the specific algorithm employed. For theorem search, we utilize `SCIP` for linear programming, and for SOS, we employ `MOSEK` to solve the SDP problem.
>
> ### Proof template
> For natural language proof, we directy ask LLM with latex format input, here is one example:
> Let $x,y$ and $z$ be positive real numbers such that $- x - y - z + x y + x z + y z = 0$, prove that $$1 + x + y + z \geq 4 x y z$$
>
> ### Code availability
> The codebase is now ready and will be released upon publication.
>
> # Concerns
> ## Method potential
> We claim that our method holds significant potential for automated algebraic inequality proving, primarily due to its **scalability** in design.
>
> **Decomposition and search** are our foundational principles, with neuro-symbolic integration and self-improvement as features that can be extended to general inequality proving. This principle is not limited to proving only polynomial or rational function inequalities, but can be generalized to a wide range of inequalities beyond the current scope of Olympiad-style problems. For example, our decomposition approach closely resembles the methodology for solving linear matrix inequalities, supporting the generalization of our framework to this area, potentially with advanced algorithms such as PIETOOLS.
>
> To further illustrate this potential, let us consider an example of an integral inequality—an area we are actively investigating as a promising direction for future work.
>
> Let  $f(0) = 0$, $f(x) \in C^1[0,1] $. Consider the integral inequality
>
> $$
> \int_0^1 (5 - x^2) f'(x)^2\,dx \geq (\pi^2 + 2) \int_0^1 f(x)^2\,dx
> $$
>
> which can be established via SOS:
>
> $$
> LHS-RHS=4\int_0^1 \left( f'(x) - \frac{\pi f(x)}{2\tan\frac{\pi x}{2}} \right)^2 dx+ \int_0^1 (1-x^2) \left( f'(x) - \frac{f(x)}{x} \right)^2 dx \geq 0
> $$
>
> Finally, we wish to avoid making overreaching claims that our method will fundamentally advance general mathematical reasoning or development, as mathematical reasoning itself is both profound and extensive. Nonetheless, we emphasize that **algebraic inequality proving** remains a vast and essential domain, with frequent applications in mathematics and theoretical physics.
>
>
> ## Self-improvement performance
> About self-improvement performance, although the ablation study shows that the self-improvement ability only solved 8 additional problems, this does not imply that its contribution is limited.
>
> (1) Solving 8 more problems is actually a substantial improvement, as these are difficult cases considering that this is the incremental gain after all other methods have been applied. Basic methods can already solve the easy problems, so solving a few more hard problems demonstrates a significant advancement.
>
> (2) The effectiveness of IneqSearch also depends on the problems in the dataset as illustrated in Section 3.3. With a larger dataset, the benefits of self-improvement would be greater.

---

> > ### Comment · Reviewer_Hdro · 2025-08-04
> >
> > Thank you for your response. Most of my concerns have been well addressed.
> >
> > I would like to clarify Question 3: as I understand it, the key features, aside from the symbolic solvers, are (1) the LLM's role in transforming the problem and (2) the library's self-improvement mechanism. Therefore, I suggest the figure also illustrates how the LLM transforms a problem (for instance, into a new problem format) or how the library evolves (for instance, by adding a new theorem, $TH_{11}$) during an iteration.
> >
> > Additionally, I remain concerned about the performance of the library's self-improvement. While I agree that library learning can be effective, the new lemmas included in the library appear to be too specific and may not generalize to other problems. If this is the case, this limitation should be carefully discussed in the paper.
> >
> > Regarding the code release, I understand that a URL cannot be provided in the rebuttal. However, it would be helpful to include an explanation for why the code was not released upon submission.

---

> ### Author Response · Authors · 2025-08-05
>
> ## About Figure 1
> For Figure 1, your understanding is correct, we mainly highlight these features to better illustrate our framework. Thank you for pointing out the clarity issue, we will revise the figure to improve its quality.
>
> ## About theorem generalization
>
> Your point is important and is actually a direction for our future work. Currently, although we can generate a series of theorems from a single problem by adding rules, these theorems are still quite specific in form. Using LLMs to generalize learned theorems is a promising approach and could enable broader application. Our work is actively developing along this direction. We discuss this further in the Discussion section:
>
> > A promising research direction involves utilizing LLMs to extract generalizable patterns from specific problems. This approach aligns with established mathematical methodology: observation of patterns, formulation of conjectures, and subsequent verification. For example, consider the following IMO problem: for positive real numbers $a, b, c, d$ such that $ab + bc + cd + da = 1$, show that $\frac{a^3}{b+c+d} + \frac{b^3}{c+d+a} + \frac{c^3}{d+a+b} + \frac{d^3}{a+b+c} \geq \frac{1}{3}$. This problem can be extended to a broader class of inequalities, such as $\sum_{i}^n \frac{x_i^3}{s-x_i} \geq \frac{4k^2}{(n-1)n}$ where $s = \sum_i x_i$ and $k^2=\sum_{i<j} x_ix_j$. The key insight is recognizing that denominators of the form $s - x_i$ and the cyclic condition $ab+bc+cd+da = 1$ can be abstracted to the setting $k^2 = \sum_{i<j} x_ix_j$, representing an elementary symmetric polynomial of degree 2.  Maintaining homogeneity during this abstraction process is essential. Such generalizations require pattern recognition and analytical reasoning capabilities well-suited to LLMs' strengths.
>
>
> ## About code release
> We didn't release the code at submission because the project is a bit complex and we hadn't finished organizing it. The codebase is about 1MB of pure Python script. We will make the code open-source to ensure reproducibility.

---

> > ### Comment · Reviewer_Hdro · 2025-08-07
> >
> > Thank you for your response! It gives me a better understanding of this work. I would like to keep my positive score.

---

### Official Review · Reviewer_fNdM · 2025-06-29

**Clarity:** 4
**Significance:** 2
**Originality:** 2
**Rating:** 3
**Confidence:** 4

**Summary:**

The paper introduces IneqSearch, a hybrid reasoning system for automated proofs of multivariate polynomial inequalities common in math Olympiads. The symbolic core first normalizes the problem, then attempts to rewrite the inequality as a sum of elementary cyclic polynomials (ECPD) or a sum of squares (SOS). When this direct route fails, an LLM agent suggests transformations derived from standard lemmas (e.g., Cauchy-Schwarz) that create alternate but equivalent targets. A linear-programming search picks coefficients that express the transformed inequality as a non-negative combination of known theorems. Every successful proof is logged, checked, and appended to the theorem base, letting the system solve progressively harder problems in subsequent rounds.

**Questions:**

•	The solver currently targets multivariate polynomial inequalities in positive variables. What prevents it from handling rational functions, sign-changing variables, or transcendental terms?

•	Have you identified theoretical or practical obstacles to extending the approach beyond Olympiad-style problems?


•	How sensitive is performance to the ordering in which new theorems are added?

•	What are the main failure modes of the system (e.g. failure to find decompositions, incorrect inductive transformations), and how might these be mitigated in future work?

•	How large is the final theorem database after training, and what are its computational or memory implications for scaling to broader problem sets?

•	Are there plans to release the code and trained theorem database to enable reproducibility and benchmarking by the community?

**Ethical Concerns:**

["NO or VERY MINOR ethics concerns only"]

**Final Justification:**

The authors rebuttal addressed some of my concerns partially that led to my increasing my rating for the paper.

**Limitations:**

•	Evaluation is restricted to polynomial inequalities with positive variables under Olympiad assumptions; generality to broader mathematical domains is untested.

•	Iterative learning risks compounding errors if incorrect proofs are integrated into the database; the safeguards to prevent this are not fully detailed.

•	Current symbolic solver methods may not generalize to inequalities involving transcendental or higher-order functions.

**Quality:**

3

**Strengths And Weaknesses:**

Strengths:

•	Strong empirical results (however, it needs to be evaluated on more datasets): IneqSearch achieves a 78.3% success rate on INEQ-437 and solves 100% of the MO-INT-20 dataset, outperforming both pure LLM-based systems and prior hybrid methods like AIPS and LIPS.

•	Comprehensive evaluation, including ablation studies demonstrating the contribution of each system component (symbolic solver, inductive agent, iterative learning).

•	Ablation study demonstrating the contribution of each system component (symbolic solver, inductive agent, iterative learning).

Weaknesses:

•	Inconsistent model coverage in Table 1. All three settings (natural-language, Lean 4, and formal verification) should be benchmarked on the same set of LLMs, yet the formal column reports results for only two models, one of which (“4o-mini”) never appears in the natural-language experiment, making direct cross-modal comparisons impossible.

•	Lack of open-weight baselines. The study omits freely available models, hindering reproducibility and preventing the community from validating or extending the findings without paid API access. (also using deterministic sampling or setting a seed)

•	Order-dependent accuracy. Reported success rates assume the exact sequence of previously posed questions; performance could drop under a different interaction history, but this sensitivity is neither analyzed nor controlled.


•	Missing comparison with retrieval-augmented generation (RAG). Given that the task involves answering from a fixed corpus, a RAG baseline is essential to demonstrate the advantage of the proposed method over standard retrieval + generation pipelines.

•	Unavailable code and data. The appendix does not include the implementation or evaluation scripts, so the reported results cannot be reproduced or independently verified.

•	While the iterative learning mechanism is promising, its scalability to larger or more diverse domains is untested; results are currently limited to algebraic inequalities within Olympiad settings.


•	No analysis of failure modes by problem category (e.g. symmetric vs asymmetric inequalities, degree complexity), which would provide deeper insights for model improvement.

•	Lack of statistical significance testing or variance reporting for experimental results, though the absolute performance gains are substantial.

•	Crucial implementation details, such as the LP search procedure, SDP solver configurations, and LLM prompting strategies, are missing, making it difficult for readers to reproduce or build upon the work. Additionally, the dataset used is limited.

Quality:

The paper presents a well-motivated methodology grounded in mathematical practice, with good empirical validation and rigorous decomposition algorithms.

Clarity:

The writing is generally clear and well-organized. Diagrams (e.g. Figures 1–3) effectively illustrate the pipeline and proof construction.

Significance:

This work is significant for automated mathematical reasoning, particularly in algebraic inequality proofs, an area of interest for both AI for mathematics and AGI evaluation benchmarks. The hybrid design could inspire extensions to broader formal domains.

Originality:

The paper frames inequality proving as a decomposition-based search integrated with LLM-guided inductive exploration, and in its self-improving theorem acquisition paradigm to enrich the database of theorems.

---

> ### Author Rebuttal · Authors · 2025-07-31
>
> Thanks for your review.
>
> Even though DeepMind recently used Gemini (with natural language) to win a gold medal in the IMO, our results show that LLMs still face significant limitations in Olympiad inequality proofs. Algebraic inequality proving is exceptionally challenging because it involves navigating an infinite reasoning space and requires both rigorous deductive reasoning and constructive proof techniques. Combining symbolic computation with LLMs enables two complementary modes of thought—rigorous deduction and creative exploration—making this hybrid approach especially powerful for tackling inequalities.
>
> # Takeaways
>
> 1. Relying solely on LLMs to generate natural or formal proofs remains challenging. Our work demonstrates that integrating symbolic solvers with LLMs enables hybrid reasoning, resulting in substantial performance improvements. T
>
> 2. The decomposition and search framework offers a promising direction for automated inequality proving. By leveraging LLMs for implementation within this framework, our results establish a practical path forward for general automated inequality proving.
>
> # Question response
> ## Question 1
> IneqSearch is **Not** limited to multivariate polynomial inequalities (please see dataset.pdf in the supplementary materials). It can also solve inequalities involving radicals, exponents, logarithms, rational functions, and more. These types of inequalities are very common in mathematical Olympiads, so our system faces no fundamental obstacles in handling such problems.
>
> As illustrated in the paper, cases involving negative variables can generally be addressed with similar methods, although they may require additional case analysis or domain partitioning. If real variables are needed, our system does not encounter any fundamental barriers.
>
> ## Question 2
>
> As we place no restriction on the analytic form of functions, **theoretically** we can solve all inequalities by decomposition, provided that the relevant theorems exist in our theorem database. We also do not assume any initial axioms in the theorem base, any necessary results from other areas can be incorporated as needed, enabling IneqSearch to continually learn and expand its capabilities.
>
> We conduct experiments on Olympiad inequalities due to the abundance of resources and the relative simplicity of their forms. However, our approach is fully extensible to broader classes of algebraic inequalities. We claim that our method holds significant potential for automated algebraic inequality proving, primarily due to its **scalability** in design.
>
> **Decomposition and search** are our foundational principles, with neuro-symbolic integration and self-improvement as features that can be extended to general inequality proving. This principle is not limited to proving only polynomial or rational function inequalities, but can be generalized to a wide range of inequalities beyond the current scope of Olympiad-style problems. For example, our decomposition approach closely resembles the methodology for solving linear matrix inequalities, supporting the generalization of our framework to this area, potentially with advanced algorithms such as PIETOOLS.
>
> To further illustrate this potential, let us consider an example of an integral inequality—an area we are actively investigating as a promising direction for future work:
>
> Let  $f(0) = 0$, $f(x) \in C^1[0,1] $. Consider the integral inequality
>
> $$
> \int_0^1 (5 - x^2) f'(x)^2\,dx \geq (\pi^2 + 2) \int_0^1 f(x)^2\,dx
> $$
>
> which can be established via SOS:
>
> $$
> \text{LHS} - \text{RHS} = 4\int_0^1 \left( f'(x) - \frac{\pi f(x)}{2\tan\frac{\pi x}{2}} \right)^2 dx + \int_0^1 (1-x^2) \left( f'(x) - \frac{f(x)}{x} \right)^2 dx \geq 0
> $$
>
>
>
> ## Question 3
> The reported accuracy is **Not** order-dependent. As illustrated in the original paper, we conduct experiments by round (Section 4.4):
>
> > In each round, IneqSearch traverses the entire benchmark and attempts to prove each inequality. The theorem base is updated once after completing the round, ensuring that any newly discovered results become available as tools for subsequent iterations.
>
> This procedure guarantees that the accuracy does not depend on the order in which inequalities are processed.
>
> ## Question 4
>
> When a problem fails to be proved, it means IneqSearch has tried all available methods—including finding decompositions and applying various inductive transformations—but none succeeded. Therefore, we are unable to point out a specific stage where the failure occurred; rather, all proof attempts were unsuccessful.
>
> Instead, we can provide an analysis of the error cases, which was somewhat lacking in the original paper. To offer a more intuitive understanding of the difficult cases in the benchmark, we analyze the failed instances to better illustrate the types of challenging problems encountered by IneqSearch.
>
> ### Uncommon Equality Conditions
> As discussed in the Limitation section of the Appendix, a necessary condition for a successful search $F = \sum_i F_i$ is that all equality cases must be satisfied for each $F_i$. Some problems, however, have uncommon equality conditions. For example,
>
> $$
> \frac{256}{3125} \geq x z^4 + x^4 y + y^4 z
> $$
>
> with $x + y + z = 1$ attains equality when $(x, y, z) = (4/5, 1/5, 0)$ under cyclic permutation. Another illustrative example is
>
> $$
> \frac{1}{81} \geq \frac{xyz}{(x+1)(x+y)(y+z)(z+16)},
> $$
>
> where equality holds at $(x, y, z) = (2, 4, 8)$.
>
> Solving such problems often requires more sophisticated, constructive algebraic proofs.
>
> ### Non-directly Usable Assumptions
> A significant proportion of problems involve assumptions that are either non-standard or couple the variables in ways that prevent direct normalization or substitution techniques. For example, constraints such as
>
> $$
> \frac{1}{x+1} + \frac{1}{y+1} + \frac{1}{z+1} = 2
> $$
>
> or
>
> $$
> x + y + z + 1 = 4xyz
> $$
>
> make it challenging for solvers to apply classical inequality lemmas directly.
>
> The key to addressing such problems often lies in transforming the given condition into a more manageable form, rather than directly substituting it into the inequality.
>
> ### Non-homogeneity
> IneqSearch relies on homogeneity information to decompose inequalities. However, the majority of unsolved inequalities in this benchmark are non-homogeneous.
>
> For example, consider
>
> $$
> 8 + 2x^2 + 2y^2 + 2z^2 + xyz \geq 5x + 5y + 5z,
> $$
>
> where the left-hand side contains constant, quadratic, and cubic terms, while the right-hand side is purely linear. The lack of a uniform degree structure complicates both analytic and algorithmic approaches.
>
> Such non-homogeneous structures often arise from the underlying construction of inequalities involving constants, such as the simple case $x^2 - 2x + 1 \geq 0$, which contains non-homogeneous terms. The algebraic construction involving constant terms remains a significant obstacle for our symbolic solvers.
>
> ## Question 5
>
> After learning on our benchmark, we ultimately obtain a total of 171 theorems.
> In our implementation, theorem search primarily consists of three steps: formalizing the equation, applying SymPy’s `solve_undetermined_coeffs` function, and solving a linear program using `SCIP`.
>
> The algorithmic complexity of these components is as follows:
>
> - Equation formalization: This step involves symbolic manipulation and pattern matching, which can be computationally intensive depending on the structure of the input but is typically polynomial in the size of the expression.
> - Solve_undetermined_coeffs: Solving for undetermined coefficients often reduces to solving a system of linear equations, which generally has cubic complexity in the number of variables (i.e., $\mathcal{O}(n^3)$).
> - Linear programming with SCIP: SCIP solves linear programs in polynomial time for most practical cases, but the worst-case complexity depends on the specific algorithm used and the size of the problem.
>
> In practice, the main computational bottleneck is the equation formalization step, with a typical runtime between 30 and 40 seconds. If the theorem base becomes very large, we may need to optimize this step further. The memory cost is not significant, as the algorithm filters applicable theorems during equation formalization—typically fewer than 10 theorems.
>
>
> ## Question 6
> The codebase is now ready and will be released upon publication.
>
> # Concerns about LLM comparison
> We are not very familiar with the specific meaning of the RAG baseline (standard retrieval + generation pipelines) in this context. Our work focuses on inequality proving and general inequality search, with access to a theorem base. Are you suggesting that we should instruct the LLM to retrieve relevant theorems and combine them to construct a proof, similar to how proofs are written in Lean? If so, we are concerned that this approach may not be ideal, as it does not allow for automatic verification or leverage solver-based deduction, and generating proofs directly in natural language may be more natural and straightforward for our setting. We would appreciate it if you could clarify how this task could be naturally formulated as a RAG problem in our case.
>
> For the open-weight models, we additionally include Qwen3-32B and Llama3.3-70B for natural language proof. On INEQ-437, Qwen3 solves 59 problems and Llama3.3 solves 48 problems, corresponding to solving rates of 13.5% and 10.9%, respectively.

---

> > ### Comment · Reviewer_fNdM · 2025-08-05
> >
> > Thank you for spending the time trying to address many of my concerns, though many weaknesses are really not addressed.
> >
> > Table 1 shows inconsistent model coverage across the three experimental settings (natural language, Lean 4, and formal verification); notably, the formal verification column only includes results for two models—one of which ("4o-mini") is absent from the natural-language experiments—making direct cross-modal comparisons infeasible. Thank you for adding a couple of open weight results in your rebuttal.
> >
> > The lack of a retrieval-augmented generation (RAG) baseline is another significant omission, especially given the corpus-constrained nature of the task, where RAG systems are commonly applied. For clarification, all the new theorems created can be locally stored.
> >
> > While the proposed iterative learning mechanism is promising, its evaluation is restricted to algebraic inequalities in math Olympiad problems, with no evidence of scalability to broader domains. Moreover, the paper does not include an error analysis broken down by problem category—such as symmetric versus asymmetric inequalities or complexity of polynomial degree—which would offer valuable insights into model behavior and failure modes. Thus, broader testing is required which is a critical miss.
> >
> > Finally, the absence of statistical significance testing and variance reporting raises concerns about the reliability of the reported gains, particularly in the absence of standard deviation or confidence intervals. You have given specific examples in your rebuttal but that is not an analysis.
> >
> > Thus, I will keep my current rating. I believe this work requires more evaluation before it can be published.

---

> ### Author Response · Authors · 2025-08-06
>
> Thank you for your detailed feedback and for engaging with our rebuttal. We would like to address your concluding comment that "many weaknesses are really not addressed." We respectfully believe this stems from a misunderstanding of our work’s core contribution, problem scope, and evaluation standards in automated theorem proving.
>
> Your concerns mainly fall into two categories:
> - Evaluation Methodology: Model coverage, missing RAG baseline, lack of statistical metrics.
> - Contribution Scope: Scalability and absence of some failure analyses.
>
> Our main argument is that these are not unaddressed weaknesses, but rather reflect a **misalignment** between standard empirical ML evaluation frameworks and our task, which is **deterministic, search-based, and logic-driven**.
>
> We focus specifically on Olympiad-level inequality proving—a recognized **grand challenge** where past methods have struggled. The value of our work lies in its depth and success in this difficult subdomain.
>
> Below, we clarify our choices and their alignment with standards in our field:
>
> ---
>
> - Model Coverage
>
> Table 1 benchmarks IneqSearch against SOTA methods within their respective categories, not for direct cross-modal comparison.
>
> Fair SOTA Comparison: In the formal proof setting, LIPS [1] is the main SOTA baseline. For fairness, we used the same language model as LIPS to evaluate IneqSearch.
>
> Uniqueness: IneqSearch uniquely generates pure algebraic proofs verifiable by symbolic computation. No other method produces this output, so direct comparison in this aspect was not possible.
>
> - RAG Baseline
>
> A RAG baseline is **not appropriate** for our core mechanism. As LeanDojo [2] and others show, RAG excels at **premise selection**, while IneqSearch is a **symbolic solver** performing deterministic, algebraic decomposition—a structured search, not a retrieval task. Formulating it as RAG would not yield a meaningful baseline.
>
> - Statistical Metrics
>
> These metrics do **not apply** to our evaluation context. Theorem proving is deterministic: for any problem, a proof is either found or not. This binary evaluation is standard in ATP. As in recent work on inequalities [1,3], geometry [7], and general ATP [2,4,5,6], statistical metrics are not reported due to the deterministic nature of the task.
>
> - Scalability
>
> We have demonstrated in the rebuttal how our framework generalizes within inequality proving, including to integral inequalities, illustrating its scalability.
>
> - Error analysis
>
> To address this limitation, we provide a supplementary error analysis broken down by problem category. Specifically, Table 5 presents the results for symmetric versus non-symmetric inequalities, as well as for polynomial and non-polynomial problems:
>
> **Table 5.** Analysis of key properties
>
> | Status    | Symmetric | Non-Symmetric | Polynomial | Not Polynomial |
> |-----------|:---------:|:-------------:|:----------:|:--------------:|
> | Solved    |   260     |      82       |    113     |      229       |
> | Unsolved  |    60     |      35       |     19     |       76       |
> | Total |   320     |     117       |    132     |      305       |
>
> ---
>
> **Table 6.** Analysis of polynomial inequalities by degree
>
> | Degree | Solved | Unsolved | Total |
> |:------:|:------:|:--------:|:-----:|
> | 1      |   1    |    0     |   1   |
> | 2      |  13    |    3     |  16   |
> | 3      |  45    |    4     |  49   |
> | 4      |  26    |    5     |  31   |
> | 5      |   2    |    3     |   5   |
> | 6      |  24    |    0     |  24   |
> | 7      |   1    |    0     |   1   |
> | 8      |   0    |    2     |   2   |
> | 9      |   0    |    1     |   1   |
> | 15     |   1    |    1     |   2   |
>
>
> We hope these clarifications comprehensively address the remaining concerns. We are confident that IneqSearch represents a substantial and well-validated contribution to automated mathematical reasoning and kindly ask for a reconsideration of our work.
>
> # Reference
> [1] Wei, C., Sun, M., & Wang, W. (2024). Proving olympiad algebraic inequalities without human demonstrations. *NeurIPS 37*, 82811-82822.
>
> [2] Yang, K. et al. (2023). LeanDojo: Theorem Proving with Retrieval-Augmented Language Models. *NeurIPS Datasets and Benchmarks*.
>
> [3] Li, Z. et al. (2025). Proving Olympiad Inequalities by Synergizing LLMs and Symbolic Reasoning. *ICLR 2025*.
>
> [4] Ren, Z. Z. et al. (2025). Deepseek-prover-v2: Advancing formal mathematical reasoning via reinforcement learning for subgoal decomposition. *arXiv:2504.21801*.
>
> [5] Lin, Y. et al. (2025). Goedel-prover: A frontier model for open-source automated theorem proving. *arXiv:2502.07640*.
>
> [6] Li, Z. et al. (2024). A Survey on Deep Learning for Theorem Proving. *1st Conf. on Language Modeling*.
>
> [7] Trinh, T. H. et al. (2024). Solving olympiad geometry without human demonstrations. *Nature*, 625, 476-482.

---

> > ### Comment · Reviewer_fNdM · 2025-08-08
> >
> > Thank you for your detailed engagement. Based on your detailed explanation, I will increase my rating by a point.

---

> ### Author Response · Authors · 2025-08-08
>
> Thank you again for your valuable time and positive re-evaluation. We are very encouraged by your willingness to raise our score and truly appreciate that our explanations have been effective.
>
> Our ultimate goal is to ensure our work is not just acceptable, but a clear and strong contribution that you can support with confidence. We feel we are on the cusp of addressing all major concerns, thanks in large part to your insightful guidance.
>
> To help us cross that final hurdle, we would be immensely grateful if you could share what you see as the most significant remaining reservation or the key aspect that currently prevents you from fully recommending acceptance.
>
> Understanding this final piece of the puzzle would be incredibly valuable to us. We are committed to resolving any remaining issues and hopefully earning your enthusiastic support for our paper.
>
> Thank you once more for your constructive and open dialogue.

---

### Official Review · Reviewer_kEQx · 2025-06-30

**Clarity:** 2
**Significance:** 3
**Originality:** 2
**Rating:** 4
**Confidence:** 4

**Summary:**

The paper introduces IneqSearch, a novel hybrid reasoning system designed to automatically prove complex Olympiad-level inequalities. Similar to LIPS system in ICLR'25, this paper combines symbolic computation for rigorous deductive reasoning with large language models (LLMs) for innovative, constructive proof exploration. A key feature of IneqSearch is its iterative learning mechanism, allowing it to continuously expand its theorem database with newly proven results, thereby enhancing its capabilities over time without human intervention. Experimental results demonstrate that IneqSearch significantly outperforms existing methods by successfully solving a high percentage of challenging inequality problems, showcasing the power of integrating symbolic and neural approaches for mathematical reasoning.

**Questions:**

1.  Formal Proof System Details and Soundness: Please elaborate on the specific formal proof system and verification mechanism used by IneqSearch to "ensure soundness by constructing verifiable proofs" and guarantee that "every theorem in T has been rigorously validated". How does this internal verification process compare in rigor and automation to the "automatically verified" correctness of formal proofs in Lean4 mentioned for baseline solutions?
    *   Criteria for Score Increase: A clear and detailed explanation of the underlying formal logic or automated verification framework (beyond just human readability of steps) that guarantees the soundness of IneqSearch's proofs would significantly increase the confidence in the system's fundamental "Quality" and "Clarity".

2.  Computational Budget and Component Performance: What are the typical computational complexity and runtime (latency) characteristics of the Elementary Cyclic Polynomial Decomposition (ECPD) and Sum of Squares (SOS) decomposition algorithms, especially considering SOS is formulated as a semidefinite programming problem? Furthermore, what is the maximum or average time budget allowed per problem for IneqSearch to achieve the state-of-the-art results presented in Table 1? For instance, a comparison point like LIPS specifies a maximum of 1.5 hours per problem.
    *   Criteria for Score Increase: Providing quantitative details on the computational cost of these core symbolic components and a clear time budget per problem for the reported results would greatly enhance the "Clarity" and "Quality" of the experimental evaluation, allowing for a more fair and comprehensive comparison of computational resources with other systems.

3.  LLM Error Handling, Backtracking, and Strategy Management: Given that Large Language Models (LLMs) can exhibit "critical consistency issues" and "hallucinations", how does IneqSearch robustly manage errors introduced by the LLM's "constructive proof exploration"? Specifically, what are the mechanisms for backtracking when an LLM-proposed transformation or exploration direction (e.g., "Inequality Adaption" or "Target Shift" in Figure 2) proves incorrect or unprovable, and how does the system balance or prioritize these various strategies during the iterative trial-and-error process?
    *   Criteria for Score Increase: A detailed explanation of the system's error recovery, backtracking, and adaptive strategy selection mechanisms would significantly improve the understanding of its "Quality" and robustness in handling the inherent unpredictability of LLM-driven exploration, which is crucial for its overall effectiveness as a hybrid system.

4.  Unique Contribution of ECPD to Iterative Learning: Given that Table 2 shows a relatively modest gain (8 problems) from the 'Iterative Learning' component, and external research [1] suggests limitations in LLM-assisted symbolic library learning, could the authors elaborate on the *unique contribution* of ECPD within the iterative learning mechanism? Specifically, does ECPD's role extend beyond initial problem-solving to enhance the *quality* or *reusability* of theorems for subsequent learning, and are there further ablation studies that could demonstrate this?
    *   Criteria for Score Increase: A clear explanation or additional experimental evidence that elucidates how ECPD, as a symbolic decomposition method, specifically contributes to the robustness, generalizability, or practical utility of the theorems acquired through iterative learning, thereby differentiating IneqSearch's approach from limitations observed in other LLM-assisted symbolic library learning studies, would enhance the understanding of its "Originality & Significance" and "Quality".

[1] Berlot-Attwell, Ian, Frank Rudzicz, and Xujie Si. "LLM Library Learning Fails: A LEGO-Prover Case Study." arXiv preprint arXiv:2504.03048 (2025).

**Ethical Concerns:**

["NO or VERY MINOR ethics concerns only"]

**Final Justification:**

Most of my concerns are addressed except for some detailed implementations. Given author's commitment on open source, I would like to raise my score.

**Limitations:**

I generally like this paper, especially on its significant results, the introduction of ECPD, and the surprising effect of iterative learning mechanism. All my concerns are mainly around the clarity of this paper, and I hope this can be addressed in the future version.

**Quality:**

3

**Strengths And Weaknesses:**

Strengths:

*   Quality & Significance: The paper introduces IneqSearch, a hybrid reasoning system that significantly outperforms existing methods on Olympiad-level inequalities. It successfully proves 342 out of 437 problems and achieves a 100% success rate on the challenging MO-INT-20 dataset. The system's strong performance on high-difficulty problems, which often require multi-step scaling and are non-symmetric or non-homogeneous, highlights its effectiveness where symbolic solvers and LLMs typically fail. An ablation study clearly demonstrates the complementary nature and individual contributions of its components, showing that each added feature boosts performance.
*   Originality & Significance: IneqSearch's core strength and innovation lies in the ECPD and SOS methods. It reformulates inequality proving as a structured search problem focused on decomposing expressions into non-negative components, a strategy inspired by formal mathematical practice, which further enables effective iterative learning mechanism that autonomously incorporates newly proven results into its theorem database, allowing for continuous knowledge acquisition and enhanced capabilities without human intervention.

Weaknesses:

*   Clarity & Quality (Formal Proof System & Soundness): While the paper states that IneqSearch "ensures soundness by constructing verifiable proofs" and that "every theorem in T has been rigorously validated, guaranteeing the system’s soundness", the specific details of its formal proof system and verification mechanism are not fully elaborated in the provided excerpts. This leaves ambiguity regarding how its correctness is guaranteed compared to systems like Lean, which have automatically verifiable formal proofs.
*   Clarity & Quality (Computational Budget & Comparison Fairness): The paper mentions that "No GPU was used in the experiments" and "All computations can be executed on a standard personal computer". However, key details regarding the computational complexity and exact runtime of the decomposition algorithms like ECPD and SOS are not clearly specified. Without these details, it is difficult to fully assess the fairness of comparisons with other systems, such as LIPS, especially concerning the computational budget required for the main results. The paper also states that the code is not yet ready for publication.
*   Clarity & Quality (Trial-and-Error Process): The inductive agent proposes "multiple possible choices at each step", implying a trial-and-error approach. However, the paper does not explicitly specify the technical details of this trial-and-error process, including any backtracking mechanisms or how the cost associated with failed attempts is factored into the reported computational budget or performance metrics.
*   Clarity (Theoretical Scope of Decomposition): While the system effectively uses ECPD and SOS for polynomial non-negativity and preprocessing converts fractional expressions to numerator-only forms, the paper does not clearly specify the theoretical completeness of these methods for *all* types of inequalities, particularly in relation to the broader mathematical context of Hilbert's Seventeenth Problem, which deals with *rational functions*. The system primarily operates on polynomials after preprocessing, which may limit its applicability to certain classes of inequalities compared to a universal theoretical claim.
*   Clarity (Limited Demonstrated Impact and Justification of Iterative Learning's Contribution, particularly ECPD's Role):
The paper identifies iterative learning as a distinguishing feature and a key contribution, enhancing IneqSearch's proving capability and scalability by autonomously incorporating solved inequalities into its knowledge base. However, the ablation study in Table 2 shows that the "Iterative Learning" mechanism is responsible for solving only an additional 8 problems (from 334 to 342). This comparatively modest gain raises questions regarding the practical significance of this component's overall impact. Furthermore, while Elementary Cyclic Polynomial Decomposition (ECPD) is mentioned as a core method for "representing and manipulating expressions" in inequality proofs, its specific and unique contribution to *enhancing iterative learning* (beyond its direct application in the solver) is not clearly demonstrated to be substantial or to significantly overcome limitations that other research (e.g., [1]) might suggest for symbolic library learning.

[1] Berlot-Attwell, Ian, Frank Rudzicz, and Xujie Si. "LLM Library Learning Fails: A LEGO-Prover Case Study." arXiv preprint arXiv:2504.03048 (2025).

---

> ### Author Rebuttal · Authors · 2025-07-31
>
> ## 1
> IneqSearch generates **pure algebraic proofs**, whose soundness can be guaranteed and verified by symbolic computation engines. While our proofs are not formally verified by an axiomatic logic system, they are validated through explicit algebraic computation. Our implementation is based on SymPy and Mathematica; below we provide a detailed explanation.
>
> Given an input expression $F$ to IneqSearch, all possible algebraic manipulations can be divided into three categories:
>
> - **Rewriting**: $F = \sum_i F_i$, including ECPD transformations, SOS, theorem search, etc.
> - **Inequality Transformation**: $F \geq 0 \Leftrightarrow G \geq 0$, including removing denominators, removing radicals, etc.
> - **Scaling**: $F \geq G$, used only during LLM exploration.
>
> The first two types of manipulations are mathematically equivalent.
>
> The third operation, scaling, is carefully constrained to ensure correctness. For example, when instructing the LLM to apply the Cauchy-Schwarz inequality, $(\sum A^2)(\sum B^2) \geq (\sum AB)^2$, we directly provide $A$ according to the specific problem and let the LLM construct $B$. Regardless of the choice of $B$, the resulting scaling **always preserves validity**, though it may introduce over-scaling. This procedure is conceptually similar to constructing `tactics` in Lean, as implemented in LIPS.
>
> To enhance the reliability of our implementation, we employ both symbolic and numerical checks.
>
> For the final algebraic proof $F \geq \sum F_i$, if there is no scaling, _i.e._, the LLM is not involved and $F = \sum F_i$, we directly verify the equality by symbolic computation. In our approach, each $F_i$ is either a square or a previously established theorem, both of which are non-negative, so the overall proof is guaranteed to be correct. If scaling is involved, we additionally verify that the inequality construction is correct by including assertion statements to ensure symbolic correctness during the construction process.
>
> Additionally, after removing denominators or radicals, or when scaling occurs, we perform numerical verification by sampling variable values. While not mathematically necessary, this step serves as an extra safeguard against potential implementation errors.
>
> ## 2
> **Table 3.** Runtime analysis of the proof workflow on INEQ-437.
> | Module                    | Process                      | Avg. Runtime (s) | Time Budget (s) |
> |---------------------------|------------------------------|------------------|-----------------|
> |                           | ECPD Transform               | 0.6              | 30              |
> |                           | Theorem Search               | 41.6             | 240             |
> |                           | SOS                          | 0.4              | 30              |
> | Solver                    | **Total**                    | **12.3**         | **300**         |
> |                           |                              |                  |                 |
> |                           | Equality Condition           | 17.3             | 180             |
> |                           | Target Shift                 | 19.1             | 180             |
> |                           | AM-GM Ineq. Expl.            | 32.6             | 180             |
> |                           | Cauchy-Schwarz Ineq. Expl.   | 27.3             | 180             |
> |                           | Hölder's Ineq. Expl.         | 41.0             | 180             |
> |                           | Jensen's Ineq. Expl.         | 34.4             | 180             |
> |                           | Tangent Line Ineq. Expl.     | 37.9             | 180             |
> | Inductive Agent (o3-mini) | **Total**                    | **65.8**         | **300**         |
> |                           |                              |                  |                 |
> | **Overall**               |                              | **24.1**         | **900**         |
>
> ---
>
> **Table 4.** Comparison of solving time and success rate.
> | Methods                  | Avg. Runtime (s) | Success Rate |
> |--------------------------|------------------|--------------|
> | OpenAI o3-mini           | 63               | 35.2%        |
> | LIPS w/ 120 min budget   | 5187             | 62.5%        |
> | LIPS w/ 90 min budget    | 2946             | 50.3%        |
> | IneqSearch w/o LLM       | 12               | 66.8%        |
> | IneqSearch w/ o3-mini    | 24               | 78.3%        |
>
>
> ---
>
> Table 3 summarizes the average runtime and time budget for the main proving steps of IneqSearch with o3-mini on our INEQ-437 benchmark. Only the most time-consuming steps are shown, and steps may not be strictly sequential in practice. Reported runtimes are per single execution, though steps may be called multiple times per proof. For successfully solved problems, IneqSearch requires only 12.3 seconds per problem on average, demonstrating **high efficiency**.
>
> Table 4 shows the average solving time for successful proofs, comparing our method to LIPS and o3-mini. Our approach achieves over a 100× speedup compared to LIPS. However, this direct runtime comparison is not fully fair algorithmically, since most of LIPS's time cost comes from repeated LLM calls.
>
> Regarding computational complexity, IneqSearch relies on several computational tools, and the precise complexity depends on the specific algorithm employed. For theorem search, we utilize `SCIP` for linear programming, and for SOS, we employ `MOSEK` to solve the SDP problem.
>
> ## 3
> #### Prompt Template
>
> Here, we present the prompt template used to automate the application of the Cauchy-Schwarz inequality with the LLM.
>
> > We are solving an inequality problem. Please do not use the code interpreter.
> >
> > The inequality is:
> > `{S_str}`
> >
> > We have the following important information:
> >
> > 1. Equality holds for: `{cond_str}`
> > 2. Assumption condition: `{assump_str}`
> >
> > Now, we want to use the Cauchy-Schwarz inequality to scale $S$, so that we can obtain an expression which is easier to handle.
> > The term indices are:
> > `{term_dict}`
> >
> > Recall that the Cauchy-Schwarz inequality is:
> >
> > $$
> > \left(\sum_{i=1}^n A_i\right) \left(\sum_{i=1}^n B_i\right) \geq \left(\sum_{i=1}^n C_i\right)^2
> > $$
> >
> > or
> >
> > $$
> > \sum_{i=1}^n C_i \leq \sqrt{\left(\sum_{i=1}^n A_i\right) \left(\sum_{i=1}^n B_i\right)},
> > $$
> >
> > where $C_i = \sqrt{A_i B_i}$
> >
> > **Instructions:**
> >
> > 1. Identify the terms that are suitable for the Cauchy-Schwarz inequality, and return their term indices.
> > 2. Determine the role of these terms as $A_i$ or $C_i$ based on their coefficients: $A_i$ for positive, $C_i$ for negative.
> >     - (1) If the role is $A_i$, use the first form of the given Cauchy-Schwarz inequality; otherwise, use the second.
> >     - (2) Ensure that the role ($A_i$ or $C_i$) is included among the terms of $S$.
> > 3. Design $B_i$ so that the other role ($A_i$ or $C_i$) is easy to handle.
> > 4. If the Cauchy-Schwarz inequality is applicable, return the corresponding term indices as a list of integers.
> >
> > Provide your analysis and calculation steps, and summarize your answer in a JSON. The JSON format begins with <code>```json</code> as shown below.
> > Each term should be a LaTeX string without the `$` symbol, and the product sign `*` should be explicitly written between parentheses in the expressions:
> >
> > ```json
> > {
> >   "applicable": true or false,
> >   "role": "A_i" or "C_i",
> >   "term_numbers": [num1, num2, ...] corresponding to the indices of $A_1/C_1$, $A_2/C_2$, ...
> >   "B_1": latex formula corresponding to $A_1$ or $C_1$,
> >   "B_2": latex formula corresponding to $A_2$ or $C_2$,
> >   ...
> > }
> > ```
>
> For each input inequality, we decompose it into individual algebraic terms, assign each term a unique index, and represent them in LaTeX format as a dictionary (`term_dict`).
> For example, for the inequality $a^2 + b^2 + c^2 \geq ab + bc + ca$, the `term_dict` would be
> `{1: "a^2", 2: "b^2", 3: "c^2", 4: "-ab", 5: "-bc", 6: "-ca"}`.
>
> #### Post-processing
>
> In the model response, we extract the relevant terms from `term_dict` according to the `term_numbers` in the JSON, and combine them with the extracted $B_i$ expressions to automatically assemble a standard Cauchy-Schwarz inequality.
>
> ## 4
> Different from library learning as in LEGO-Prover, our self-improvement ability is developed from scratch: we do not require any answer annotations or external supervision. Instead, our system relies on discovering new theorems for difficult cases.
>
> When proving an inequality by decomposition, _i.e._, $F = \sum_i F_i$, the core idea is to find components $F_i$ that can "support" $F$. For example, suppose $F = F_1 + F_2$ with $F_1 \geq 0$ and $F_2 \geq 0$. It may be that $F \geq F_1$ holds on the interval $(0,1)$, while $F \geq F_2$ holds on $(1, +\infty)$. Only their complementarity allows us to prove $F \geq 0$ on the entire domain.
>
> About self-improvement performance, although the ablation study shows that the self-improvement ability only solved 8 additional problems, this does not imply that its contribution is limited.
>
> (1) Solving 8 more problems is actually a substantial improvement, as these are difficult cases considering that this is the incremental gain after all other methods have been applied. Basic methods can already solve the easy problems, so solving a few more hard problems demonstrates a significant advancement.
>
> (2) The effectiveness of IneqSearch also depends on the problems in the dataset as illustrated in Section 3.3. With a larger dataset, the benefits of self-improvement would be greater.
>
> ECPD is a representation method especially effective for Olympiad-style inequalities. For more general inequalities without symmetric or cyclic structure, however, ECPD offers no particular advantage. This is not a limitation, but rather reflects the framework's flexibility—ECPD is just a convenient and efficient option. IneqSearch does not heavily rely on ECPD.

---

> ### Comment · Reviewer_kEQx · 2025-08-06
>
> Thank you for the detailed rebuttal. I now have a better understanding of the technical details behind the system. However, I still find several critical issues unresolved. Therefore, I will maintain my current score, leaning toward a negative recommendation. Below are my follow-up comments and questions:
>
> 1. On the Underlying Verification Mechanism
>
> While I agree that expressing the target expression as a sum of squares (SOS) is sufficient to guarantee non-negativity, I believe some important details in the verification process remain unclear:
>
> (1) Algebraic transformations without scaling
>
> You mention that in the absence of scaling, the correctness of transformations can be verified symbolically. Given the wide variety of possible transformations (e.g., removing radicals, substitutions, removing denominators), can all such manipulations be soundly verified using symbolic computation? If so, could the authors specify which tools (SymPy, Mathematica, etc.) and what exact implementations or algebra systems are used?
>
> (2) Assertions under scaling
>
> You note that when scaling is involved, correctness is ensured via assertion statements. Could you clarify what these assertions entail? For instance, in Figure 3, if a scaling is introduced under certain assumptions, how are these assumptions checked and validated in the backend system?
>
> (3) Use of numerical sampling
>
> You mention numerical sampling as an extra check. However, this is not a valid substitute for formal proof, and could introduce a major unfair advantage for your system. For comparison, the LIPS system uses cylindrical algebraic decomposition (CAD) as a numerical guide, but ultimately still requires formal proof checking in Lean. If IneqSearch treats numerical checks as proof, this would significantly inflate your system’s apparent accuracy compared to fully verified baselines.
>
> 2. On Runtime and Search Dynamics
>
> In Table 3, several elementary inequalities (e.g., AM-GM, Cauchy-Schwarz) are reported to take ~30 seconds with o3-mini, while the average total runtime per problem is ~24 seconds. This suggests minimal backtracking or retries during LLM exploration.
>
> * Does this imply that o3-mini is already capable of selecting and executing the correct scaling strategies in one pass?
>
> * If so, how do you reconcile this result with prior work (e.g., Sheng et al. [1]) which found that o3's unreliable performance in inequality reasoning?
>
> 3. On the Prompt Template and Absence of Backtracking
>
> Thank you for providing the Cauchy-Schwarz prompt template. However, I do not see any mechanisms related to backtracking or retrying failed strategies. Does this mean your system currently does not implement backtracking?
>
> While you do mention issues like over-scaling, it is unclear whether your system can recover from such situations via LLM-guided search.
>
> 4. On Self-Improvement and Decomposition Strategy
>
> I appreciate the authors’ clarification on how decomposition contributes to solving harder problems. I agree that solving 8 additional cases in difficult settings is a meaningful improvement. However, I have a few questions regarding search efficiency:
>
> * How is the search space structured and traversed when attempting to discover helpful decompositions?
>
> * Specifically, how is ECPD (Extended Cyclic Polynomial Decomposition) combined with theorem search or decomposition to improve search efficiency in complex cases?
>
> Overall, I acknowledge the strengths of the system and the improvements over prior work. However, I remain concerned about the rigor of proof validation, the lack of backtracking, and the potential inflation of success rates via numerical heuristics. I hope the authors can address these points in future revisions.
>
> [1] Sheng, Jiayi, et al. "Solving Inequality Proofs with Large Language Models." arXiv preprint arXiv:2506.07927 (2025).

---

> ### Author Response · Authors · 2025-08-07
>
> Thank you for your detailed follow-up and for providing us the opportunity to clarify these key technical aspects of our system. We appreciate your insightful questions, which help us articulate the finer points of our methodology. We address each of your concerns below.
>
> ## On the Underlying Verification Mechanism
>
> We agree that the rigor of the verification process is paramount. Our system's soundness is guaranteed by backend **symbolic engines**.
>
> (1) All non-scaling manipulations are handled by SymPy and Mathematica. Key operations are implemented and verified as follows:
>
> - **Substitutions & Simplification:**
>   The ECPD transform is implemented in Mathematica by solving the system $C_{k_1, k_2, \ldots, k_m}^{(n)} = \sum_{cyc} x_1^{k_1} x_2^{k_2} \cdots x_m^{k_m}$.
>     ```
>     (* symPolyEqs defines the relation between original and ECPD bases, vars are x_i *)
>     sol = Solve[symPolyEqs, vars, Reals]
>     varsVals = vars /. sol[[1]]
>     F_ECPD = Simplify[F>=0]
>     ```
> - **Removing Denominators:**
>   For a fractional expression $F = P/Q$, we use SymPy to separate numerator and denominator, then prove the sign of $Q$ before simplifying.
>     ```
>     P, Q = sympy.together(F).as_numer_denom()
>     ```
> - **Removing Radicals:**
>   We use SymPy's `unrad` to eliminate radicals where possible, then check the co-factors (`cov_unrad`) to ensure validity.
>     ```
>     F_unrad, cov_unrad = sympy.solvers.solvers.unrad(F)
>     ```
> - **Final End-to-End Verification:**
>   A proof is successful **if and only if** the final equality $F = \sum_i F_i$ holds symbolically, where each $F_i$ is non-negative.
>     ```
>     is_correct = sympy.simplify(F - sum(Fi)) == 0
>     ```
>   This ensures the composite proof step is correct, even if individual manipulations are not formally checked.
>
> (2)
> The "assertions" we perform are concrete symbolic checks to ensure a proposed scaling is constructed and applied correctly. For example, in Figure 3, where $F = (1 + \frac{1}{x})(1 + \frac{1}{y})(1 + \frac{1}{z}) - 64$, with $x + y + z = 1$:
>
> - After expanding $F$, we provide the LLM with a dictionary of terms {1: 1/x, 2: 1/y, 3: 1/(x*y), ...}.
> - The LLM proposes a construction (e.g., Cauchy-Schwarz application) via JSON; we assert the JSON is well-formed and that each B_i is a valid LaTeX formula.
> - The system constructs and verifies the scaling inequality programmatically. For the example:
>     $$
>     \left(\frac{1}{x} + \frac{1}{y} + \frac{1}{z} + \frac{1}{xy} + \dots\right) \left(x + y + z + xy + \dots\right) \geq (1+1+1+1+1+1+1)^2
>     $$
>   The resulting inequality is then passed to the solver. For other lemmas (e.g., Jensen's), we assert/verify convexity as needed.
>
> (3)
> Numerical sampling is **NEVER** used as a substitute for proof. It only acts as a fast heuristic filter to discard obviously false LLM suggestions or catch bugs.
>
> ## On Runtime and Search Dynamics
>
> Your observation about average runtime is correct. The overall average (24s) is lower than the Inductive Agent's (66s) because **many problems are solved directly by our symbolic solver without LLM involvement**. The 65.8s runtime reflects the harder problems that require LLM exploration.
>
> Regarding o3-mini's performance, our findings are consistent with IneqMath[1]. LLMs like o3 have "fragile deductive chains," and our system is specifically designed to mitigate this:
>
> - **Role Separation:** The LLM is a **heuristic strategist** only; rigorous deduction and verification are offloaded to the symbolic engine.
> - **Performance Consistency:** The differing o3-mini success rates (35.2% vs. 9.5%) are explained by dataset differences. Our benchmark is broader; IneqMath is focused on very hard Olympiad problems.
>
> ## On the Prompt Template and Absence of Backtracking
>
> Our prompt does not include backtracking logic. This is intentional, as **backtracking is managed by the main solver's search algorithm, not the LLM agent.**
>
> - **Retry:** One retry is allowed for superficial LLM failures.
> - **Backtracking:** Handled by our solver's best-first search; if a scaling leads to a dead-end, other nodes are explored. This is systematic and LLM-agnostic.
>
> ## On Self-Improvement and Decomposition Strategy
>
> Thank you for acknowledging the significance of solving 8 additional hard problems. The search for a decomposition $F = \sum \lambda_i f_i$ is formulated as an **LP problem**, solving for coefficients $\lambda_i \geq 0$. The search space of $f_i$ is all the theorem base.
>
> ECPD significantly improves the search for symmetric/cyclic problems by providing a **canonical basis**, dramatically pruning the search space. For example, in Figure 3, $xy + yz + zx - 63xyz + 2$ has 4 variable terms in its original form but only 2 in ECPD ($C_{1,1}^{(3)}$ and $C_{1,1,1}^{(3)}$). This reduction allows more effective theorem filtering and a much smaller, tractable LP problem.
>
> We hope these explanations resolve your concerns. We are confident in the rigor of our symbolic verification and the design of our system.

---

> > ### Comment · Reviewer_kEQx · 2025-08-07
> >
> > Thank you again for your detailed clarification. Your responses have helped me better understand several important aspects of the system design, particularly regarding the backend verification mechanism and the impact of ECPD on symbolic library learning. However, there remain some key details that I believe are either missing or insufficiently specified, which currently prevents me from increasing my overall evaluation. I would appreciate further clarification on the following points:
> >
> > (1) On Scaling Strategy Generation and Solver Verification
> >
> > You mention: “After expanding F, we provide the LLM with a dictionary of terms…” and “The system constructs and verifies the scaling inequality programmatically.” However, after expansion, many symmetric structures are lost, and the number of variable combinations can be overwhelming. Given this, how do you ensure that the LLM can still identify the correct scaling strategy? As we know, effective scaling often relies on treating certain subexpressions as holistic units—a non-trivial task.
> >
> > Additionally, you note that “The resulting inequality is then passed to the solver” and that for lemmas such as Jensen's, you “assert/verify convexity as needed.” Could you specify:
> >
> > What exact type of solver is used here?
> >
> > How do you ensure the solver execution strictly follows the assumptions of the invoked inequality (e.g., AM-GM, Jensen, tangent line method)?
> >
> > Some of these scaling strategies require both algebraic manipulation and numerical heuristics (e.g., tangent line constructions). Given your “Role Separation” design, how are such composite reasoning tasks coordinated between the LLM and the backend system?
> >
> > (2) On Runtime and Pattern Matching Efficiency
> >
> > Regarding runtime: I was not referring to the average 24s vs 66s comparison, but rather to Table 3, where operations such as “Equality Condition” and other scaling-based strategies show runtimes in the 17s–38s range. The ratio for 66s and these numbers appears surprisingly low, indicating that the o3-mini model seems unable to perform more than three to four scaling steps on most problems.
> >
> > Can you clarify whether the system avoids exhaustive enumeration of combinations when applying these strategies? Human experts often need to explore many candidate combinations due to combinatorial explosion. How does your system manage to maintain high pattern-matching efficiency on complex inequalities?
> >
> > (3) On Backtracking and Dataset Implications
> >
> > You noted that backtracking is handled by your solver's best-first search, not by the LLM itself. Is the backtracking time included in the latency numbers reported in Table 3? If so, does that imply that most Ineq-437 problems—when solved using scaling strategies—require only shallow exploration (i.e., minimal backtracking)?
> >
> > If this is the case, does it suggest that the Ineq-437 dataset might be particularly favorable for your system’s architecture? Could you also report runtime statistics on a dataset such as MO-INT-20, which presumably contains harder instances and would test your system’s ability to handle deeper backtracking and complex scaling strategies?
> >
> > Thank you again for your thorough engagement. I look forward to your clarifications.

---

> > > ### Author Response · Authors · 2025-08-08
> > >
> > > Thank you for your follow-up questions. It is a pleasure to discuss these technical details with you. We will first provide some key implementation details about our exploration process and then address each of your concerns directly.
> > >
> > > For clarity, let $\mathcal{S}$ denote our symbolic solver engine, which is the core orchestration component of the entire IneqSearch framework, and $\mathcal{M}$ denote the LLM-based inductive agent.
> > >
> > > #### Core Implementation of the Exploration Mechanism
> > > During exploration, the LLM acts as a **proposer**. In our implementation, there are 5 inequality lemmas to be explored and the process is conducted **in parallel**. Once $\mathcal{M}$ returns its proposals, the symbolic engine $\mathcal{S}$ takes over to rigorously validate and construct the resulting scaled expressions.
> > >
> > > The application of the AM-GM inequality and Hölder's inequality is quite similar to that of the Cauchy-Schwarz inequality. The construction is largely controlled and the scaling form is fixed.
> > >
> > > For lemmas like Jensen's inequality and the Tangent Line inequality, the LLM is prompted to return a structured JSON proposal. Upon receiving this, $\mathcal{S}$ executes a series of programmatic checks. **The LLM's proposal is never trusted outright.** Our instruction part in the prompt is as follows (we omit the information about $F$ given earlier):
> > >
> > > - Jensen's inequality
> > >     > ...
> > >     > Follow these steps to determine whether the original inequality is suitable for using Jensen's inequality.
> > >     > 1. Identify the function f(t) which is suitable to apply Jensen's inequality to some terms.
> > >     > 2. Analyze whether f(t) is convex or concave for t > 0.
> > >     > 3. If f(t) is convex, its coefficient should be positive; if concave, negative.
> > >     > 4. If Jensen's inequality is applicable to some terms, return the function f and corresponding terms.
> > >
> > > - Tangent Line inequality
> > >     > ...
> > >     > Follow these steps to determine whether the original inequality is suitable to use the tangent line trick.
> > >     > 1. Identify the function f(t) which is suitable to apply the tangent line trick to some terms.
> > >     > 2. Analyze whether f(t) is convex or concave for t > 0.
> > >     > 3. If f(t) is convex, its coefficient should be positive; if concave, negative.
> > >     > 4. If the tangent line trick is applicable to some terms, return the function f and corresponding terms.
> > >
> > > We ask the LLM to return the specific form of the function and corresponding variables.
> > >
> > > > Give me your analysis steps and summarize them in a Json for me.
> > > > ```json
> > > > {
> > > >     "applicable": true or false,
> > > >     "t1": latex string of a variable,
> > > >     "f(t1)": latex string of a term,
> > > >     "t2": latex string of a variable,
> > > >     "f(t2)": latex string of a term,
> > > >     ...
> > > > }
> > > > ```
> > >
> > > After collecting its proposal, the solver checks the validity of the latex format and the convexity of the proposed function. The solver then controls the construction. Here is sample code for Jensen's inequality:
> > > ``` python
> > > # Parse and validate the LLM's proposal
> > > try:
> > >     t1 = parse_latex(rsp_dict['t1'])
> > >     ft1 = parse_latex(rsp_dict['f(t1)'])
> > >     assert applicable and ineq.has(ft1)
> > > except:
> > >     return None
> > >
> > > # Generate and verify the "proof obligation" for convexity
> > > t = sp.Symbol('t', real=True, positive=True)
> > > coeff = ineq.coeff(ft1)
> > > ft1c = coeff*ft1
> > > ft2 = ft1c.subs({t1:t})
> > > diff1 = sp.diff(ft2, t)
> > > diff2 = sp.diff(diff1, t)
> > > is_convex = sp.solve_univariate_inequality(diff2>=0, v)
> > >
> > > # Construct new inequality
> > > if is_convex:
> > >     all_perms = cyclic_permutations(ineq.free_symbols)
> > >     avg_term = 0
> > >     for perm in all_perms:
> > >         j_t = ft1.subs(perm, simultaneous=True)
> > >         coeff = ineq.coeff(j_t)
> > >         j_v = t1.subs(perm, simultaneous=True)
> > >         avg_term += j_v*coeff
> > >
> > >     ineq_new += n*ft1.subs({t1:t}).subs({t:avg_term/n})
> > >     ineq_new -= ft_sum
> > >     return ineq_new
> > > ```
> > >
> > > For the Tangent Line inequality, we manually select the equal point as the contact point.
> > > ```python
> > > # Determine validity
> > > c_point = cond_equation[t1]
> > > c_dict = {t:c_point}
> > > ft3 = ft2.subs(c_dict) + diff1.subs(c_dict)*(t-c_point)
> > > is_valid = sp.solve_univariate_inequality((ft2 - ft3)>=0, v)
> > >
> > > # Construct new inequality
> > > if is_valid:
> > >     ineq_new = ineq
> > >     ft3c = ft3.subs({t:t1})
> > >     ineq_new -= ft1_sum
> > >     ineq_new += sum_cycle(ft3c)
> > > ```

---

> ### Author Response · Authors · 2025-08-08
>
> Now, we address your specific questions, starting with the role of our solver.
>
> ## On Scaling Strategy Generation and Solver Verification
> > **What exact type of solver is used here?**
>
> This is a crucial point that strikes at the heart of our contribution. The "solver" to which a new inequality is passed is not a separate, third-party tool. It is **IneqSearch itself**.
>
> The process is **recursive**. When a scaling strategy is applied, it transforms the problem into a new, often simpler, inequality. This new inequality is not handed off to a different component; instead, it is pushed back onto the main **best-first search stack** as a new sub-goal to be solved by the **entire IneqSearch framework**.
>
>
> Therefore, the LLM agent $\mathcal{M}$ is best understood as a powerful **heuristic function** that the main search algorithm $\mathcal{S}$ can call upon when it needs to expand its search space with a creative, strategic move.
>
>
> > **How do you ensure the solver execution strictly follows the assumptions?**
>
> This is handled naturally by the recursive architecture described above. When $\mathcal{S}$ validates a lemma (e.g., Jensen's), it generates **new proof obligations** (e.g., "prove $f''(x) \geq 0$"). These obligations become new sub-goals in the search tree, which must be solved by the same IneqSearch engine. The main proof branch is only allowed to proceed if all of its dependent obligations are successfully discharged.
>
> > **How does the LLM identify scaling strategies on complex expressions?**
>
> We acknowledge that our current design has a specific focus. Our philosophy is to find an **overall decomposition** of the problem. While this may miss some highly localized tricks, we believe it is a robust approach for a general-purpose solver. This is a deliberate design choice that favors global structure over micro-level manipulation.
>
>
> ## On Runtime and Pattern Matching Efficiency
>
> Your analysis of the low number of scaling steps is correct, and it points to a deliberate design choice. As shown in Figure 1, the solving pipeline is a sequence of $\mathcal{S} \rightarrow \mathcal{M} \rightarrow \mathcal{S} \rightarrow \mathcal{M}\rightarrow \dots$. We set the **maximum number of exploration calls to $\mathcal{M}$ to 2**. This is because IneqSearch is designed to rely on its **self-improvement mechanism** (learning new theorems for $\mathcal{S}$) to tackle complexity, rather than performing "heavy exploration" (deep, multi-step scaling searches) within a single proof attempt.
>
> ## On Backtracking and Dataset Implications
>
> > **Is the backtracking time included in the latency numbers reported in Table 3?**
>
> Yes. To be precise: the runtimes for specific processes (e.g., "Equality Condition") are the average times per single, successful execution. The "Total" and "Overall" runtimes are for the **entire proof process**. Therefore, **backtracking time is included in the "Total/Overall" numbers**, as it is part of the work done by the main best-first search algorithm in $\mathcal{S}$.
>
> > **Could you also report runtime statistics on a dataset such as MO-INT-20?**
>
> This is an excellent suggestion. To test our system's performance on harder instances, we have run our system on the **MO-INT-20 dataset**. We report the detailed, per-problem solving times for MO-INT-20 in **Table 5 of our revised manuscript**. These results provide a clear picture of our system's performance on more difficult problems.
>
>
> **Table 5.** Runtime (in seconds) for each problem in MO-INT-20.
>
> | Problem ID | Time (s) |
> |------------|----------|
> | 0          | 11       |
> | 1          | 13       |
> | 2          | 1        |
> | 3          | 1        |
> | 4          | 1        |
> | 5          | 195      |
> | 6          | 1        |
> | 7          | 96       |
> | 8          | 1        |
> | 9          | 3        |
> | 10         | 13       |
> | 11         | 2        |
> | 12         | 1        |
> | 13         | 1        |
> | 14         | 1        |
> | 15         | 1        |
> | 16         | 183      |
> | 17         | 1        |
> | 18         | 342      |
> | 19         | 1        |
> |           |         |
> | **Avg.**   | **43.5** |
>
>
> We hope this detailed architectural clarification and the new results fully address your concerns. Thank you again for your thorough engagement.

---

> > ### Author Response · Authors · 2025-08-09
> >
> > Dear Reviewer kEQx,
> >
> > Thank you once again for your incredibly thorough and insightful engagement with our work. Your sharp questions have been instrumental in helping us refine and clarify our core contributions.
> >
> > We just wanted to gently confirm that you have seen our latest response from August 8th. In it, we aimed to provide a comprehensive explanation for your final set of questions, particularly focusing on the recursive nature of our solver, the mechanics of the search and backtracking process, and the new runtime statistics on the MO-INT-20 dataset.
> >
> > We are hopeful that these additional details fully address your remaining concerns. We deeply appreciate the time and expertise you have dedicated to our manuscript and are, of course, available for any further clarifications.
> >
> > Thank you again for your invaluable guidance throughout this process.

---

### Official Review · Reviewer_eiiF · 2025-07-01

**Clarity:** 2
**Significance:** 2
**Originality:** 3
**Rating:** 4
**Confidence:** 4

**Summary:**

The paper introduce IneqSearch, a neuro-symbolic system, automates the proof of Olympiad inequalities using symbolic computation and LLM-based heuristic construction. The system decomposes the target inequality into a sum of non-negative expressions, which are either established theorems in a database or constructed by LLMs. Proved inequalities are then added to the database, leading to a self-improving system. The system’s performance is evaluated on two datasets, demonstrating its significant superiority over existing baselines.

**Questions:**

- It is unclear to me how the generated proofs are verified. Does IneqSearch generate proofs in Lean4?
- In the evaluation, other methods for comparison are evaluated for three shots. How does it count for one shot? For example, in LIPS, each attempt is allocated 90 minutes. Does one shot of LIPS take 90 minutes in the evaluation?

**Ethical Concerns:**

["NO or VERY MINOR ethics concerns only"]

**Final Justification:**

The authors have provided sufficient technical details, including the verification process and agent configuration, which strengthen the credibility of their contribution. However, the proposed techniques appear to be narrowly tailored to the inequality proving, limiting their generalizability to broader applications. As such, while the work is technically sound, its overall impact may be constrained by its limited scope.

**Limitations:**

While the details of the agent design are absent, the potential application of paper is limited to automated inequality proving.

**Quality:**

3

**Strengths And Weaknesses:**

Strengths:
+ The task of proving Olympiad inequalities addressed by the paper is challenging.
+ The proposed approach is mathematically well-founded.
+ The paper collects a large dataset of Olympiad inequalities, which is valuable for further development.
+ The performance of the experiment is impressive, significantly surpassing existing baselines.

Weaknesses:
- The design of the LLM-based inductive agent is not adequately presented. Given that the agent is a crucial component of this neuro-symbolic system, the paper should provide more detailed information about the agent, including the prompt and some example inputs.
- In addition to the absence of agent information, the paper requires further justification for the design of the agent. As evidenced in Table 2, the performance of the symbolic component alone is already impressive. The combination of ECPD and SOS effectively resolves 257 problems, closely approximating the performance of LIPS, which addresses 273 problems. Without additional justification, it appears that the majority of the improvement is attributed to the symbolic component, which is limited to inequality proving problems. As a result, its influence on broader neuro-symbolic systems may be reduced.

---

> ### Author Rebuttal · Authors · 2025-07-31
>
> Thanks for your review.
>
> Even though DeepMind recently used Gemini (with natural language) to win a gold medal in the IMO, our results show that LLMs still face significant limitations in Olympiad inequality proofs. Algebraic inequality proving is exceptionally challenging because it involves navigating an infinite reasoning space and requires both rigorous deductive reasoning and constructive proof techniques. Combining symbolic computation with LLMs enables two complementary modes of thought—rigorous deduction and creative exploration—making this hybrid approach especially powerful for tackling inequalities.
>
> # Takeaways
>
> 1. Relying solely on LLMs to generate natural or formal proofs remains challenging. Our work demonstrates that integrating symbolic solvers with LLMs enables hybrid reasoning, resulting in substantial performance improvements. T
>
> 2. The decomposition and search framework offers a promising direction for automated inequality proving. By leveraging LLMs for implementation within this framework, our results establish a practical path forward for general automated inequality proving.
>
> # Question response
> ## Question 1
> IneqSearch generates **pure algebraic proofs**, whose soundness can be guaranteed and verified by symbolic computation engines. While our proofs are not formally verified by an axiomatic logic system, they are validated through explicit algebraic computation. Advanced numerical algorithms (such as SOS) ultimately produce symbolic proofs, ensuring that the numerical process does not affect the algebraic correctness of the final result. Our implementation is based on SymPy and Mathematica; below we provide a detailed explanation.
>
> Given an input expression $F$ to IneqSearch, all possible algebraic manipulations can be divided into three categories:
>
> - **Rewriting**: $F = \sum_i F_i$, including ECPD transformations, SOS, theorem search, etc.
> - **Inequality Transformation**: $F \geq 0 \Leftrightarrow G \geq 0$, including removing denominators, removing radicals, etc.
> - **Scaling**: $F \geq G$, used only during LLM exploration.
>
> The first two types of manipulations are mathematically equivalent.
>
> The third operation, scaling, is carefully constrained to ensure correctness. For example, when instructing the LLM to apply the Cauchy-Schwarz inequality, $(\sum A^2)(\sum B^2) \geq (\sum AB)^2$, we directly provide $A$ according to the specific problem and let the LLM construct $B$. Regardless of the choice of $B$, the resulting scaling **always preserves validity**, though it may introduce over-scaling. This procedure is conceptually similar to constructing `tactics` in Lean, as implemented in LIPS.
>
> To enhance the reliability of our implementation, we employ both symbolic and numerical checks.
>
> For the final algebraic proof $F \geq \sum F_i$, if there is no scaling, _i.e._, the LLM is not involved and $F = \sum F_i$, we directly verify the equality by symbolic computation. In our approach, each $F_i$ is either a square or a previously established theorem, both of which are non-negative, so the overall proof is guaranteed to be correct. If scaling is involved, we additionally verify that the inequality construction is correct by including assertion statements to ensure symbolic correctness during the construction process.
>
> Additionally, after removing denominators or radicals, or when scaling occurs, we perform numerical verification by sampling variable values. While not mathematically necessary, this step serves as an extra safeguard against potential implementation errors.
>
> ## Question 2
> For LLM natural language proofs, we queried the LLM three times for each problem, generating three independent responses. If any one of these responses was correct, we considered the attempt successful. For LIPS, if it exceeded the time budget or the maximum number of iterations, the attempt was marked as failed.
> Similarly, we performed three independent runs of LIPS and considered a problem solved if at least one run succeeded.
>
>
> # Concerns
> ## Inductive Agent Configuration
> #### Prompt Template
>
> Here, we present the prompt template used to automate the application of the Cauchy-Schwarz inequality with the LLM.
> Other exploration prompts are similar and will be included in the revised version.
>
> **Prompt for Cauchy-Schwarz Inequality Exploration**
> >
> > We are solving an inequality problem. Please do not use the code interpreter.
> >
> > The inequality is:
> > `{S_str}`
> >
> > We have the following important information:
> >
> > 1. Equality holds for: `{cond_str}`
> > 2. Assumption condition: `{assump_str}`
> >
> > Now, we want to use the Cauchy-Schwarz inequality to scale $S$, so that we can obtain an expression which is easier to handle.
> > The term indices are:
> > `{term_dict}`
> >
> > Recall that the Cauchy-Schwarz inequality is:
> >
> > $$
> > \left(\sum_{i=1}^n A_i\right) \left(\sum_{i=1}^n B_i\right) \geq \left(\sum_{i=1}^n C_i\right)^2
> > $$
> >
> > or
> >
> > $$
> > \sum_{i=1}^n C_i \leq \sqrt{\left(\sum_{i=1}^n A_i\right) \left(\sum_{i=1}^n B_i\right)},
> > $$
> >
> > where $C_i = \sqrt{A_i B_i}$
> >
> > **Instructions:**
> >
> > 1. Identify the terms that are suitable for the Cauchy-Schwarz inequality, and return their term indices.
> > 2. Determine the role of these terms as $A_i$ or $C_i$ based on their coefficients: $A_i$ for positive, $C_i$ for negative.
> >     - (1) If the role is $A_i$, use the first form of the given Cauchy-Schwarz inequality; otherwise, use the second.
> >     - (2) Ensure that the role ($A_i$ or $C_i$) is included among the terms of $S$.
> > 3. Design $B_i$ so that the other role ($A_i$ or $C_i$) is easy to handle.
> > 4. If the Cauchy-Schwarz inequality is applicable, return the corresponding term indices as a list of integers.
> >
> > Provide your analysis and calculation steps, and summarize your answer in a JSON. The JSON format begins with <code>```json</code> as shown below.
> > Each term should be a LaTeX string without the `$` symbol, and the product sign `*` should be explicitly written between parentheses in the expressions:
> >
> > ```json
> > {
> >   "applicable": true or false,
> >   "role": "A_i" or "C_i",
> >   "term_numbers": [num1, num2, ...] corresponding to the indices of $A_1/C_1$, $A_2/C_2$, ...
> >   "B_1": latex formula corresponding to $A_1$ or $C_1$,
> >   "B_2": latex formula corresponding to $A_2$ or $C_2$,
> >   ...
> > }
> > ```
>
> For each input inequality, we decompose it into individual algebraic terms, assign each term a unique index, and represent them in LaTeX format as a dictionary (`term_dict`).
> For example, for the inequality $a^2 + b^2 + c^2 \geq ab + bc + ca$, the `term_dict` would be
> `{1: "a^2", 2: "b^2", 3: "c^2", 4: "-ab", 5: "-bc", 6: "-ca"}`.
> We also provide the equality condition (`cond_str`) and the assumptions (`assump_str`) as LaTeX strings in the prompt, ensuring the model has access to all relevant information during reasoning.
> The prompt instructs the model to analyze the structure of each term to determine its role in the Cauchy-Schwarz inequality as either $A_i$ or $C_i$, and to systematically design the corresponding $B_i$ for each.
>
> #### Post-processing
>
> In the model response, we extract the relevant terms from `term_dict` according to the `term_numbers` in the JSON, and combine them with the extracted $B_i$ expressions to automatically assemble a standard Cauchy-Schwarz inequality.
> For example, if the role is $A_i$, we construct:
>
> $$
> \left( \sum_{i} A_i \right)\left( \sum_{i} B_i \right) \geq \left( \sum_{i} \sqrt{A_i B_i} \right)^2.
> $$
>
> Any error encountered during this construction process (as detected by our multiple symbolic checks) is regarded as an exploration failure.
>
> ## Method potential
> We claim that our method holds significant potential for automated algebraic inequality proving, primarily due to its **scalability** in design.
>
> **Decomposition and search** are our foundational principles, with neuro-symbolic integration and self-improvement as features that can be extended to general inequality proving. This principle is not limited to proving only polynomial or rational function inequalities, but can be generalized to a wide range of inequalities beyond the current scope of Olympiad-style problems. For example, our decomposition approach closely resembles the methodology for solving linear matrix inequalities, supporting the generalization of our framework to this area, potentially with advanced algorithms such as PIETOOLS.
>
> To further illustrate this potential, let us consider an example of an integral inequality—an area we are actively investigating as a promising direction for future work.
>
> Let  $f(0) = 0$, $f(x) \in C^1[0,1] $. Consider the integral inequality
>
> $$
> \int_0^1 (5 - x^2) f'(x)^2\,dx \geq (\pi^2 + 2) \int_0^1 f(x)^2\,dx
> $$
>
> which can be established via SOS:
>
> $$
> LHS-RHS=4\int_0^1 \left( f'(x) - \frac{\pi f(x)}{2\tan\frac{\pi x}{2}} \right)^2 dx+ \int_0^1 (1-x^2) \left( f'(x) - \frac{f(x)}{x} \right)^2 dx \geq 0
> $$
>
> Finally, we wish to avoid making overreaching claims that our method will fundamentally advance general mathematical reasoning or development, as mathematical reasoning itself is both profound and extensive. Nonetheless, we emphasize that **algebraic inequality proving** remains a vast and essential domain, with frequent applications in mathematics and theoretical physics.

---

> > ### Comment · Reviewer_eiiF · 2025-08-04
> >
> > Thank you for your response. Most of my technical concerns have been addressed, and the paper is technically solid. However, the scope of the contribution, particularly regarding inequality proving, still appears limited. I have therefore updated my score to 4.

---

### Official Review · Reviewer_ZztM · 2025-07-01

**Clarity:** 3
**Significance:** 2
**Originality:** 2
**Rating:** 5
**Confidence:** 4

**Summary:**

The paper introduces IneqSearch, a hybrid system for proving Olympiad-level algebraic inequalities. A symbolic solver first tries direct decompositions and searches a growing theorem database; when that fails, an LLM-based “inductive agent” suggests transformations. Successfully proven sub-results are fed back into the database so the system “learns” over multiple passes. On the authors’ new INEQ-437 benchmark and the MO-INT-20 set, IneqSearch solves 342/437 (78 %) and 20/20 (100 %) inequalities, substantially outperforming state-of-the-art LLMs and prior hybrids such as LIPS and AIPS. The system outputs human-readable proof logs and formal Lean 4 certificates.

**Questions:**

1. How does IneqSearch perform on publicly used inequality suites such as IneqMath? [Sheng et al., 2025]
2. Could you quantify IneqSearch’s compute footprint (e.g., wall-clock time per problem, total GPU/CPU hours, number of LLM calls) and compare it directly to the baselines you evaluate against (LIPS, AIPS, pure-LLM solvers)?
3. Can you provide more details on INEQ-437's construction (e.g., difficulty distribution)?

**Ethical Concerns:**

["NO or VERY MINOR ethics concerns only"]

**Final Justification:**

The authors have addressed my concerns by reporting competitive scores on IneqMath benchmark, also updated the manuscript with compute efficiency metrics and more details about the dataset. I believe this strengthens the paper by providing more external evaluation and runtime metrics, which allows for fair comparison with other methods. The method performs well in these comparison, and therefore I raised my score.

**Limitations:**

Partially addressed - the paper contains a dedicated “Limitations” appendix, but it skips compute-resource discussion and omits the lack of external benchmarks; these should be added. Otherwise the section is adequate.

**Paper Formatting Concerns:**

no major concerns

**Quality:**

2

**Strengths And Weaknesses:**

Strengths:
* Strong empirical results
* The system has a notion of memory (theorem database) which allows for self-improvement

Weaknesses:
*  Evaluation is almost entirely on an in-house benchmark created by the authors (except MO-INT-20 which is a small set); external sets such as IneqMath are missing.
* Heavy reliance on hand-crafted ECPD/SOS + linear programming - may not generalise.
* Domain scope is narrow (inequality problems)
* The approach doesn't make proper use of stronger LLMs (the improvement from 4o-mini to o3-mini is marginal)

---

> ### Author Rebuttal · Authors · 2025-07-31
>
> Thanks for your review.
>
> Even though DeepMind recently used Gemini (with natural language) to win a gold medal in the IMO, our results show that LLMs still face significant limitations in Olympiad inequality proofs. Algebraic inequality proving is exceptionally challenging because it involves navigating an infinite reasoning space and requires both rigorous deductive reasoning and constructive proof techniques. Combining symbolic computation with LLMs enables two complementary modes of thought—rigorous deduction and creative exploration—making this hybrid approach especially powerful for tackling inequalities.
>
> # Takeaways
>
> 1. Relying solely on LLMs to generate natural or formal proofs remains challenging. Our work demonstrates that integrating symbolic solvers with LLMs enables hybrid reasoning, resulting in substantial performance improvements. T
>
> 2. The decomposition and search framework offers a promising direction for automated inequality proving. By leveraging LLMs for implementation within this framework, our results establish a practical path forward for general automated inequality proving.
>
>
> # Question response
> ## Question 1
> IneqMath is a benchmark designed to test LLMs on inequality problems, rather than focusing solely on proving inequalities. Many problems cannot be directly transformed into proof statements. For example, some require demonstrating that an inequality does not hold by providing a numerical counter-example. For evaluation, we used the dev split, which contains 100 problems in total. Among them, we manually determined that 68 problems can be converted into proof problems.  IneqSearch successfully proved 48 of these, resulting in a success rate of 70.6\%.
>
> ## Question 2
> **Table 3.** Runtime analysis of the proof workflow on INEQ-437.
> | Module                    | Process                      | Avg. Runtime (s) | Time Budget (s) |
> |---------------------------|------------------------------|------------------|-----------------|
> |                           | ECPD Transform               | 0.6              | 30              |
> |                           | Theorem Search               | 41.6             | 240             |
> |                           | SOS                          | 0.4              | 30              |
> | Solver                    | **Total**                    | **12.3**         | **300**         |
> |                           |                              |                  |                 |
> |                           | Equality Condition           | 17.3             | 180             |
> |                           | Target Shift                 | 19.1             | 180             |
> |                           | AM-GM Ineq. Expl.            | 32.6             | 180             |
> |                           | Cauchy-Schwarz Ineq. Expl.   | 27.3             | 180             |
> |                           | Hölder's Ineq. Expl.         | 41.0             | 180             |
> |                           | Jensen's Ineq. Expl.         | 34.4             | 180             |
> |                           | Tangent Line Ineq. Expl.     | 37.9             | 180             |
> | Inductive Agent (o3-mini) | **Total**                    | **65.8**         | **300**         |
> |                           |                              |                  |                 |
> | **Overall**               |                              | **24.1**         | **900**         |
>
> ---
>
>
> **Table 4.** Comparison of solving time and success rate.
> | Methods                  | Avg. Runtime (s) | Success Rate |
> |--------------------------|------------------|--------------|
> | OpenAI o3-mini           | 63               | 35.2%        |
> | LIPS w/ 120 min budget   | 5187             | 62.5%        |
> | LIPS w/ 90 min budget    | 2946             | 50.3%        |
> | IneqSearch w/o LLM       | 12               | 66.8%        |
> | IneqSearch w/ o3-mini    | 24               | 78.3%        |
>
>
> ---
>
> IneqSearch first attempts to prove each inequality using the symbolic solver. If this attempt fails, the system proceeds to call the LLM for further exploration. In our implementation, there are 5 inequality lemmas to be explored, and each inequality exploration calls the LLM once, with one additional retry if the first call is invalid. The exploration of inequality lemmas is conducted in parallel, allowing us to collect all possible valid scaled expressions after exploration.
>
> Table 3 summarizes the average runtime and time budget for the proving steps of IneqSearch with o3-mini on our INEQ-437 benchmark. Only the major time-consuming steps are listed, and the steps are not strictly sequential in the implementation. The reported runtime for each process refers to the average time per single execution, although each process may be called multiple times within a proof. For accurate measurement of actual solving time, we report the runtime for successful proofs, where IneqSearch requires only 12.3 seconds on average per problem, demonstrating **high efficiency**.
>
> Table 4 presents the average solving time for successfully solved problems and compares our method to LIPS and o3-mini. Compared to LIPS, our approach achieves more than a 100× speedup in runtime. While these results are convincing in practice, we note that this direct runtime comparison is not entirely fair from an algorithmic perspective, as the majority of the time cost in LIPS arises from repeated LLM calls. Our method is primarily solver-driven and typically requires far fewer LLM calls per problem.
>
>
> ## Question 3
> Since our benchmark does not contain any human annotations such as the number of steps, problem difficulty becomes hard to formally define. From the perspective of IneqSearch, those inequalities that can be proved by the solver at the very beginning are considered relatively easy samples (see Ablation Study). To provide a more intuitive description of the difficult cases in the benchmark, we instead analyze the failed cases to better reflect the types of challenging problems encountered by IneqSearch.
>
> ### Uncommon Equality Conditions
> As discussed in the Limitation section of the Appendix, a necessary condition for a successful search $F = \sum_i F_i$ is that all equality cases must be satisfied for each $F_i$. Some problems, however, have uncommon equality conditions. For example,
>
> $$
> \frac{256}{3125} \geq x z^4 + x^4 y + y^4 z
> $$
>
> with $x + y + z = 1$ attains equality when $(x, y, z) = (4/5, 1/5, 0)$ under cyclic permutation. Another illustrative example is
>
> $$
> \frac{1}{81} \geq \frac{xyz}{(x+1)(x+y)(y+z)(z+16)},
> $$
>
> where equality holds at $(x, y, z) = (2, 4, 8)$.
>
> Solving such problems often requires more sophisticated, constructive algebraic proofs.
>
>
> ### Non-directly Usable Assumptions
> A significant proportion of problems involve assumptions that are either non-standard or couple the variables in ways that prevent direct normalization or substitution techniques. For example, constraints such as
>
> $$
> \frac{1}{x+1} + \frac{1}{y+1} + \frac{1}{z+1} = 2
> $$
>
> or
>
> $$
> x + y + z + 1 = 4xyz
> $$
>
> make it challenging for solvers to apply classical inequality lemmas directly.
>
> The key to addressing such problems often lies in transforming the given condition into a more manageable form, rather than directly substituting it into the inequality.
>
> ### Non-homogeneity
> IneqSearch relies on homogeneity information to decompose inequalities. However, the majority of unsolved inequalities in this benchmark are non-homogeneous.
>
> For example, consider
>
> $$
> 8 + 2x^2 + 2y^2 + 2z^2 + xyz \geq 5x + 5y + 5z,
> $$
>
> where the left-hand side contains constant, quadratic, and cubic terms, while the right-hand side is purely linear. The lack of a uniform degree structure complicates both analytic and algorithmic approaches.
>
> Such non-homogeneous structures often arise from the underlying construction of inequalities involving constants, such as the simple case $x^2 - 2x + 1 \geq 0$, which contains non-homogeneous terms. The algebraic construction involving constant terms remains a significant obstacle for our symbolic solvers.
>
> # Concern about limited domain
> We claim that our method holds significant potential for automated algebraic inequality proving, primarily due to its **scalability** in design.
>
> **Decomposition and search** are our foundational principles, with neuro-symbolic integration and self-improvement as features that can be extended to general inequality proving. This principle is not limited to proving only polynomial or rational function inequalities, but can be generalized to a wide range of inequalities beyond the current scope of Olympiad-style problems. For example, our decomposition approach closely resembles the methodology for solving linear matrix inequalities, supporting the generalization of our framework to this area, potentially with advanced algorithms such as PIETOOLS.
> To further illustrate this potential, let us consider an example of an integral inequality—an area we are actively investigating as a promising direction for future work.
>
> Let  $f(0) = 0$, $f(x) \in C^1[0,1] $. Consider the integral inequality
>
> $$
> \int_0^1 (5 - x^2) f'(x)^2\,dx \geq (\pi^2 + 2) \int_0^1 f(x)^2\,dx
> $$
>
> which can be established via SOS:
>
> $$
> LHS-RHS=4\int_0^1 \left( f'(x) - \frac{\pi f(x)}{2\tan\frac{\pi x}{2}} \right)^2 dx+ \int_0^1 (1-x^2) \left( f'(x) - \frac{f(x)}{x} \right)^2 dx \geq 0
> $$
>
> Finally, we wish to avoid making overreaching claims that our method will fundamentally advance general mathematical reasoning or development, as mathematical reasoning itself is both profound and extensive.  Nonetheless, we emphasize that **algebraic inequality proving** remains a vast and essential domain, with frequent applications in mathematics and theoretical physics.

---

### Decision · Program_Chairs · 2025-09-17

**Decision:**

Accept (poster)

**Comment:**

This paper introduces IneqSearch, a neuro-symbolic system for proving Olympiad-level inequalities using structured search. The hybrid approach uses a Large Language Model for guided exploration and a symbolic solver for deductive reasoning, along with an iterative learning mechanism to expand its theorem database over time. On a new benchmark of 437 problems, IneqSearch achieved state-of-the-art performance by solving 78.3% of them, outperforming prior methods.

**(b) Strengths**

*   **Performance:** The system demonstrates impressive, state-of-the-art results on a challenging benchmark of Olympiad-level problems.
*   **Novel Methodology:** The paper presents a sound, hybrid neuro-symbolic framework.
*   **Verifiable Proofs:** Unlike purely generative models, the system produces verifiable algebraic proofs.

**(c) Weaknesses**

*   **Narrow Scope:** The method is highly specialized for Olympiad-style inequalities, and its broader applicability remains unclear.
*   **Initial Lack of Clarity:** The original submission lacked key implementation details, a runtime analysis, and comparisons on standard benchmarks, which initially raised concerns about reproducibility.

The recommendation is to **accept**. The paper makes a significant contribution by achieving outstanding results on a very challenging task. While the initial submission had weaknesses, the authors' comprehensive and diligent responses during the discussion period successfully addressed reviewer's concerns

**(e) Summary of Discussion and Rebuttal**

The discussion period was highly effective. Key points raised by reviewers included the need for evaluation on public benchmarks, clarification on the proof soundness mechanism, and a detailed runtime analysis. The authors addressed these points thoroughly by:

*   Providing new, strong results on the public IneqMath benchmark.
*   Explaining in detail how backend symbolic engines (SymPy/Mathematica) guarantee proof soundness, clarifying that numerical checks are only used as pre-filters.
*   Adding comprehensive analyses of runtime and computational costs.